# Training-Conditional Coverage Bounds under Covariate Shift

**Mehrdad Pournaderi**  *m.pournaderi@utah.edu*
*University of Utah*

**Yu Xiang**  *yxiang@fau.edu*
*Florida Atlantic University*

**Reviewed on OpenReview:** *https://openreview.net/forum?id=F6hHT3qWxT*

## Abstract

Conformal prediction methodology has recently been extended to the covariate shift setting, where the distribution of covariates differs between training and test data. While existing results ensure that the prediction sets from these methods achieve marginal coverage above a nominal level, their coverage rate conditional on the training dataset—referred to as training-conditional coverage—remains unexplored. In this paper, we address this gap by deriving upper bounds on the tail of the training-conditional coverage distribution, offering probably approximately correct (PAC) guarantees for these methods. Our results characterize the reliability of the prediction sets in terms of the severity of distributional changes and the size of the training dataset.

## 1 Introduction

Conformal prediction is a framework for constructing *distribution-free* and *model-agnostic* predictive confidence regions under the exchangeability assumption for the training and test samples (Vovk et al., 2005) (also see Shafer and Vovk (2008); Vovk et al. (2009); Vovk (2012)). Let $((X_1, Y_1), (X_2, Y_2), \ldots, (X_n, Y_n), (X_{\text{test}}, Y_{\text{test}}))$ denote a tuple of exchangeable data points, consisting of a training sequence of $n$ samples $\mathbf{D}_n := ((X_1, Y_1), \ldots, (X_n, Y_n)) \in (\mathcal{X} \times \mathcal{Y})^n$ and one test sample $(X_{\text{test}}, Y_{\text{test}}) \in \mathcal{X} \times \mathcal{Y}$. For a fixed error rate level $\alpha$, the conformal prediction framework, provides a prediction set $\hat{C}_\alpha(X_{\text{test}})$ for $Y_{\text{test}}$ that satisfies

$$\mathbb{P}\left(Y_{\text{test}} \in \hat{C}_\alpha(X_{\text{test}})\right) \geq 1 - \alpha, \tag{1}$$

where $\hat{C}_\alpha : \mathcal{X} \to 2^{\mathcal{Y}}$ is a data-dependent map. This type of guarantee is referred to as *marginal* coverage, as it is averaged over both the training and test data.

One natural direction to stronger results is to provide a coverage guarantee *conditional* on the test data point $X_{\text{test}}$, i.e.,

$$\mathbb{P}\left(Y_{\text{test}} \in \hat{\mathcal{C}}_\alpha(X_{\text{test}})|X_{\text{test}}\right) \geq 1 - \alpha.$$

However, when $X_{\text{test}}$ has a continuous distribution, it has been shown in Vovk (2012); Foygel Barber et al. (2021); Lei and Wasserman (2014) that it is *impossible* to obtain (non-trivial) distribution-free prediction regions $\hat{\mathcal{C}}_\alpha(x)$ in the finite-sample regime; relaxed versions of this type of guarantee have been extensively studied (see Jung et al. (2022); Gibbs et al. (2023); Vovk et al. (2003) and references therein). As a different approach to stronger guarantees, several results (e.g., Vovk (2012); Bian and Barber (2023)) have been reported on the *training-conditional* guarantee by conditioning on $\mathbf{D}_n$, which is also more appealing than the marginal guarantee as can be seen below. Define the following miscoverage rate as a function of the training data,

$$P_e(\mathbf{D}_n) := \mathbb{P}\left(Y_{\text{test}} \notin \hat{\mathcal{C}}_\alpha(X_{\text{test}})|\mathbf{D}_n\right).$$

Note that the marginal coverage in (1) is equivalent to $\mathbb{E}[P_e(\mathbf{D}_n)] \leq \alpha$. The training-conditional guarantees concern the concentration of the conditional error rate below the nominal level $\alpha$ and they have the following form, for some small $\delta > 0$,

$$\mathbb{P}\left(P_e(\mathbf{D}_n) \geq \alpha\right) \leq \delta$$

or its asymptotic variants. Roughly speaking, this guarantee means that the $(1 - \alpha)$-level coverage lower bounds hold for a *generic* dataset.

Recently, Barber et al. (2021) proposed modified versions of jackknife and cross validation (CV), namely *jackknife+* and *CV+*, which can be used to compute conformal prediction sets. For the $K$-fold CV+ with $\ell$ samples in each fold, the training-conditional coverage bound

$$\mathbb{P}\left(P_e(\mathbf{D}_n) \geq 2\alpha + \sqrt{2\log(K/\delta)/\ell}\right) \leq \delta \tag{2}$$

is established in Bian and Barber (2023). Additionally, Liang and Barber (2023) proposed training-conditional coverage bounds for jackknife+ and full conformal prediction sets under the assumption that the training algorithm is symmetric and satisfies certain stability conditions (see Section 4 for more details). In this line of research, samples are assumed to be i.i.d., which is not only exchangeable but also ergodic and admits some nice concentration properties. However, this assumption can be violated in the application. In particular, the input data distribution during deployment can differ from the distribution observed during training. This phenomenon is called *distribution shift* and it is a crucial problem in trustworthy machine learning (see Section 2.3). In this regard, split and full conformal prediction methods as the central methods for distribution-free uncertainty quantification of black-box models have recently been extended to handle a popular type of distribution shift called *covariate shift* (Tibshirani et al., 2019). In the covariate shift setting, the distribution of covariates in the test data differs from the one observed in the training data, but, the conditional distribution of the response given the features remains the same across the training and test populations. A similar extension has been made for the jackknife+ method Prinster et al. (2022).

Despite these significant progress on handling distribution shift, they primarily focus on marginal coverage. Although the weighted conformal prediction methods (proposed in Tibshirani et al. (2019) and Prinster et al. (2022)) are guaranteed to keep the marginal coverage rate above the nominal level, they often reduce the concentration of the training-conditional coverage rate $\mathbb{P}(Y_{\text{test}} \in \hat{C}_\alpha(X_{\text{test}})|\mathbf{D}_n)$ (i.e., coverage rate conditional on the training data $\mathbf{D}_n$) around the nominal level $1 - \alpha$. In particular, the weighting scheme leads to heavier tails for the training-conditional coverage rate. This phenomenon is illustrated in Figure 1, where ordinary and weighted conformal methods with 80% target coverage rate ($\alpha = 0.2$) are used for constructing prediction intervals under exchangeability and distribution shift settings, respectively. The distribution shift is introduced artificially via resampling according to an exponential tilt as in Section 2.3 from Tibshirani et al. (2019). It can be observed that the tails of the training conditional coverage get heavier when the weighting scheme is used to handle the distribution shift. In this paper, we explore the quality of the weighted conformal prediction sets by computing upper bounds on the tails of the training-conditional coverage distribution, quantifying the relationship between the tail behavior and the extent of distribution change. In other words, we examine the efficiency of weighted conformal prediction methods for a generic training dataset under distribution shift.

We present concentration bounds of the training-conditional coverage for weighted jackknife+ (JAW) (Prinster et al., 2022), full, and split conformal methods. Regarding the training algorithm, no assumption is made for the split conformal method. However, full conformal and jackknife+ methods are analyzed under the assumption of *uniform stability* as explained below. Let $\hat{\mu}_{\mathbf{D}_n}$ denote the predictor function estimated using the training data $\mathbf{D}_n$. A training algorithm is called uniformly stable if,

$$\|\hat{\mu}_{\mathbf{D}_n} - \hat{\mu}_{\mathbf{D}'_n}\|_\infty \leq \beta \tag{3}$$

with $\beta = O(1/n)$ for any two datasets $(\mathbf{D}_n, \mathbf{D}'_n)$ differing in one (training) data sample (Bousquet and Elisseeff, 2002). This is a stronger notion of algorithmic stability than the $(m, n)$-stability assumed in Liang and Barber (2023). Nevertheless, uniform stability is satisfied by a class of regression models known as RKHS regression (Bousquet and Elisseeff, 2002), i.e., regularized empirical risk minimization over an RKHS (Paulsen and Raghupathi, 2016; Schölkopf and Smola, 2002). Examples of reproducing kernel Hilbert spaces are certain Sobolev spaces of smooth functions (Wahba, 1990).

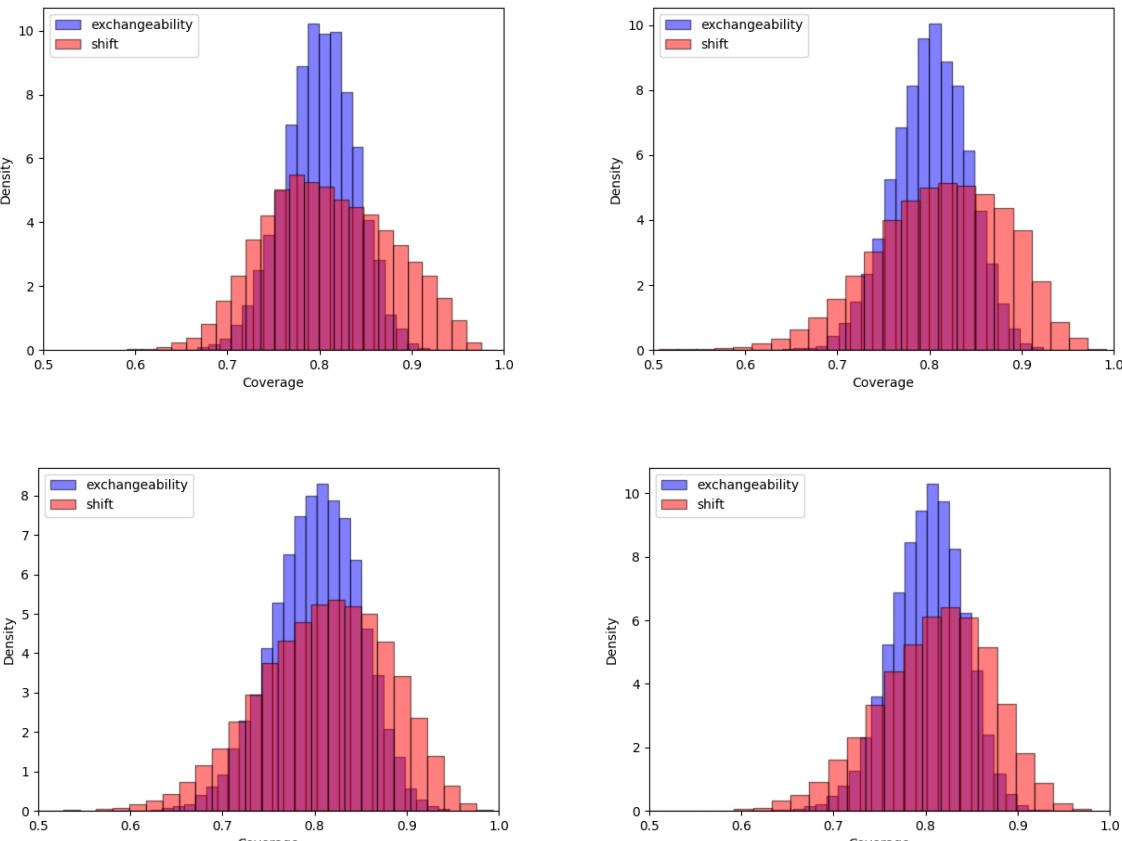

Figure 1: Histograms of the training-conditional coverage rates are presented for four datasets from the UCI Machine Learning Repository: Wine Quality (top left), Abalone (top right), Concrete Compressive Strength (bottom left), and Combined Cycle Power Plant (bottom right). See Appendix E for details of this simulation study.

## 2 Background and Related Work

### 2.1 Full conformal and split conformal

Define the shorthand $\mathbf{D}_n \cup (x, y) := ((X_1, Y_1), (X_2, Y_2), \ldots, (X_n, Y_n), (x, y))$ and let $\hat{\mu}_{(x,y)} := T(\mathbf{D}_n \cup (x, y))$ denote the regression function obtained by running the training algorithm $T$ on $\mathbf{D}_n \cup (x, y)$. Define the score function $s(x', y'; \mu) := f(\mu(x'), y')$ via some arbitrary (measurable) cost function $f : \mathcal{Y} \times \mathcal{Y} \to \mathbb{R}$ and predictor function $\mu : \mathcal{X} \to \mathcal{Y}$. For instance, $s(x', y'; \mu) = |y' - \mu(x')|$ when $f(y, y') = |y - y'|$. For the data multiset $\mathcal{D}_n := \{(X_i, Y_i) \in \mathcal{X} \times \mathcal{Y} : i \in [n]\}$ where $[n] := \{1, 2, ..., n\}$, define

$$\mathcal{S}(\mathcal{D}_n; \mu) := \{s(x', y'; \mu) : (x', y') \in \mathcal{D}_n\}.$$

Observe that if $T$ is symmetric, i.e., $T(\mathbf{D})$ is invariant under permutations of the elements of $\mathbf{D}$ for any data tuple $\mathbf{D}$, then the elements of $\mathcal{S}(\mathcal{D}_n \cup (X_{\text{test}}, Y_{\text{test}}); \hat{\mu}_{(X_{\text{test}}, Y_{\text{test}})})$ are exchangeable. Therefore,

$$\mathbb{P}\left(s(X_{\text{test}}, Y_{\text{test}}; \hat{\mu}_{(X_{\text{test}}, Y_{\text{test}})}) \leq \hat{F}^{-1}_{\mathcal{S}(\mathcal{D}_n \cup (X_{\text{test}}, Y_{\text{test}}); \hat{\mu}_{(X_{\text{test}}, Y_{\text{test}})})}(1 - \alpha)\right) \geq 1 - \alpha,$$

where $\hat{F}^{-1}_S(\cdot)$ denotes the *empirical* quantile function of $S$. Thus,

$$\mathbb{P}\left(Y_{\text{test}} \in \hat{C}^{\text{full}}_\alpha(X_{\text{test}})\right) \geq 1 - \alpha,$$

where the following confidence region is referred to as *full conformal* in the literature

$$\hat{C}_\alpha^{\text{full}}(x) = \{y : s(x, y; \hat{\mu}_{(x,y)}) \leq \hat{F}_{\mathcal{S}(\mathcal{D}_n \cup (x,y); \hat{\mu}_{(x,y)})}^{-1}(1 - \alpha)\}$$

$$\subseteq \{y : s(x, y; \hat{\mu}_{(x,y)}) \leq \hat{F}_{\mathcal{S}(\mathcal{D}_n; \hat{\mu}_{(x,y)}) \cup \{\infty\}}^{-1}(1 - \alpha)\}, \tag{4}$$

It is well-known that this approach can be computationally intensive when $\mathcal{Y} = \mathbb{R}$ since to find out whether $y \in \hat{C}_\alpha^{\text{full}}(x)$ one needs to train the model with the dataset including $(x, y)$. One simple way to alleviate this issue is to split the data into training and calibration datasets, namely $\mathcal{D}_n = \mathcal{D}^{\text{train}} \cup \mathcal{D}^{\text{cal}}$. Without loss of generality (when splitting), we assume $\mathbf{D}^{\text{train}} = ((X_1, Y_1), \ldots, (X_{n-m}, Y_{n-m}))$ and $\mathbf{D}^{\text{cal}} = ((X_{n-m+1}, Y_{n-m+1}), \ldots, (X_n, Y_n))$. In the split conformal method, one first finds the regression predictor $\hat{\mu} = T(\mathbf{D}^{\text{train}})$ and treats $\hat{\mu}$ as fixed. Note that the elements of $\mathcal{S}(\mathcal{D}^{\text{cal}} \cup (X_{\text{test}}, Y_{\text{test}}); \hat{\mu})$ are exchangeable. Hence, we get

$$\mathbb{P}\left(s(X_{\text{test}}, Y_{\text{test}}; \hat{\mu}) \leq \hat{F}_{\mathcal{S}(\mathcal{D}^{\text{cal}} \cup (X_{\text{test}}, Y_{\text{test}}); \hat{\mu})}^{-1}(1 - \alpha)\right) \geq 1 - \alpha.$$

Hence, we have $\mathbb{P}(Y_{\text{test}} \in \hat{C}_\alpha^{\text{split}}(X_{\text{test}})) \geq 1 - \alpha$ for

$$\hat{C}_\alpha^{\text{split}}(x) = \left\{y : s(x, y; \hat{\mu}) \leq \hat{F}_{\mathcal{S}(\mathcal{D}^{\text{cal}}; \hat{\mu}) \cup \{\infty\}}^{-1}(1 - \alpha)\right\} \tag{5}$$

$$\supseteq \left\{y : s(x, y; \hat{\mu}) \leq \hat{F}_{\mathcal{S}(\mathcal{D}^{\text{cal}} \cup (x,y); \hat{\mu})}^{-1}(1 - \alpha)\right\}.$$

## 2.2 Jackknife+

Although the split-conformal approach resolves the computational efficiency problem of the full conformal method, it is somewhat inefficient in using the data and may not be useful in situations where the number of samples is limited. A heuristic alternative has long been known in the literature, namely, jackknife or leave-one-out cross-validation that can provide a compromise between the full conformal and split conformal methods. In particular,

$$\hat{C}_\alpha^{\text{J}}(x) = \{y : s(x, y; \hat{\mu}) \leq \hat{F}_{\mathcal{S}^{\text{J}}}^{-1}(1 - \alpha)\},$$

where $\hat{\mu} = T(\mathbf{D}_n)$, $\mathcal{S}^{\text{J}} := \{s(X_i, Y_i; \hat{\mu}^{-i}) : 1 \leq i \leq n\}$ and $\hat{\mu}^{-i} := T(((X_1, Y_1), \ldots, (X_{i-1}, Y_{i-1}), (X_{i+1}, Y_{i+1}), \ldots, (X_n, Y_n)))$. Despite its effectiveness, no general finite-sample guarantees are known for jackknife. Recently, Barber et al. (2021) proposed jackknife+, a modified version of the jackknife for $\mathcal{Y} = \mathbb{R}$ and $f(y, y') = |y - y'|$, and established $(1 - 2\alpha)$ finite-sample coverage lower bound for it. Let

$$\mathcal{S}^-(x) = \{\hat{\mu}^{-i}(x) - |Y_i - \hat{\mu}^{-i}(X_i)| : i \in [n]\} \cup \{-\infty\},$$
$$\mathcal{S}^+(x) = \{\hat{\mu}^{-i}(x) + |Y_i - \hat{\mu}^{-i}(X_i)| : i \in [n]\} \cup \{\infty\}.$$

The jackknife+ prediction interval is defined as

$$\hat{C}_\alpha^{\text{J+}}(x) = [\hat{F}_{\mathcal{S}^-(x)}^{-1}(\alpha), \ \hat{F}_{\mathcal{S}^+(x)}^{-1}(1 - \alpha)].$$

In the same paper, an $\epsilon$-inflated version of the jackknife+

$$\hat{C}_\alpha^{\text{J+},\epsilon}(x) = [\hat{F}_{\mathcal{S}^-(x)}^{-1}(\alpha) - \epsilon, \ \hat{F}_{\mathcal{S}^+(x)}^{-1}(1 - \alpha) + \epsilon]. \tag{6}$$

is proposed which has $1 - \alpha - 4\sqrt{\nu}$ coverage lower bound (instead of $1 - 2\alpha$), if the training procedure satisfies

$$\max_{i \in [n]} \mathbb{P}(|\hat{\mu}(X_{\text{test}}) - \hat{\mu}^{-i}(X_{\text{test}})| > \epsilon) < \nu.$$

Also, the jackknife+ has been generalized to CV+ for $K$-fold cross-validation, and $(1 - 2\alpha - \sqrt{2/n})$ coverage lower bound is established.

### 2.3 Conformal prediction under distribution shift

Covariate shift concerns the setting when the covariate distribution changes between training and test data, while the conditionals remain the same. Specifically, we have

$$Z_i := (X_i, Y_i) \overset{\text{i.i.d.}}{\sim} P_X \times P_{Y|X} =: P \quad \text{training data;}$$
$$Z_{\text{test}} := (X_{\text{test}}, Y_{\text{test}}) \sim Q_X \times P_{Y|X} =: Q \quad \text{test data.}$$

This notion has been extensively studied in machine learning (e.g., see Sugiyama et al. (2007); Quinonero-Candela et al. (2008); Sugiyama and Kawanabe (2012); Wen et al. (2014); Reddi et al. (2015); Chen et al. (2016) and the references therein).

To handle the covariate shift, a weighted version of conformal prediction was first proposed in the seminal work by Tibshirani et al. (2019). The key assumption is that the likelihood ratio $dQ/dP = dQ_X/dP_X$, needs to be known; we follow the same assumption in this work. This reweighting scheme has been extended to various settings in conformal prediction Podkopaev and Ramdas (2021); Lei and Candès (2021); Fannjianga et al.; Guan (2023). The general domain adaption problem has also been an active area from a causal perspective (e.g., Zhang et al. (2013a); Peters et al. (2016); Gong et al. (2016); Chen and Bühlmann (2021); Du and Xiang (2023)).

The weighted procedures require computing a weight associated with each sample,

$$w_i = \frac{\frac{dQ}{dP}(Z_i)}{\frac{dQ}{dP}(Z_{\text{test}}) + \sum_{i \in \mathcal{I}} \frac{dQ}{dP}(Z_i)}, \; i \in \mathcal{I} \quad \text{and} \quad w_{\text{test}} = \frac{\frac{dQ}{dP}(Z_{\text{test}})}{\frac{dQ}{dP}(Z_{\text{test}}) + \sum_{i \in \mathcal{I}} \frac{dQ}{dP}(Z_i)}$$

where $\mathcal{I} = [n]$ for full conformal and jackknife+, and $\mathcal{I} \subset [n]$ for the split conformal method which corresponds to the calibration (or hold-out) dataset. In particular, to generalize the full and split conformal methods to distribution shift setting, according to Tibshirani et al. (2019) one needs to replace the empirical CDF used in construction of (4) and (5)

$$\hat{F}_{\mathcal{S}}(t) = \frac{1}{n+1} \sum_{s \in \mathcal{S}} \mathbf{1}\{s \le t\}, \quad \mathcal{S} = \{s(Z_i) : i \in \mathcal{I}\} \cup \{\infty\}$$

by the weighted version

$$F_{Q,|\mathcal{I}|}(t) = w_{\text{test}} \mathbf{1}\{t = \infty\} + \sum_{i \in \mathcal{I}} w_i \mathbf{1}\{s(Z_i) \le t\}.$$

Similarly, to extend the jackknife+ method, Prinster et al. (2022) proposes replacing $\hat{F}_{\mathcal{S}^-}$ and $\hat{F}_{\mathcal{S}^+}$ by their corresponding weighted versions where $w_{\text{test}}$ is associated to $-\infty$ and $\infty$, respectively.

## 3 Conditional Coverage Guarantees

### 3.1 Split conformal

The training-conditional coverage guarantee for split conformal method under the i.i.d. setting has been studied in Vovk (2012); Bian and Barber (2023). The proof in this setting relies on the Dvoretzky–Kiefer–Wolfowitz (DKW) inequality. The following theorem concerns training-conditional coverage guarantee under covariate shift, where the extension is made by developing a weighted version of the DKW inequality. Let $\mathbf{D}_n = (Z_1, \ldots, Z_n) \in (\mathcal{X} \times \mathcal{Y})^n$ denote an $n$-tuple of data points containing both the training and calibration data sets. Define the training conditional probability of error as

$$P_e^{\text{split}}(\mathbf{D}_n) := \mathbb{P}\left(Y_{\text{test}} \notin \hat{C}_\alpha^{\text{split}}(X_{\text{test}}) \middle| \mathbf{D}_n\right). \tag{7}$$

**Theorem 1** *Let $m < n$ denote the size of the calibration data set. Assume that $Q$ is absolutely continuous with respect to $P$ ($Q \ll P$) and $dQ/dP \le B < \infty$. Then,*

$$\mathbb{P}\left(P_e^{split}(\mathbf{D}_n) > \alpha + \left(\sqrt{2B \log(4/\delta)} + 2C\right)\sqrt{\frac{B}{m}}\right) \le \delta.$$

*for all $\delta > 0$, where $C > 0$ is a universal constant. The probability is taken with respect to $P$ since each entry in $\mathbf{D}_n$ follows $P$.*

**Corollary 1** *Relaxing the assumption $dQ/dP \leq B$ to $\|dQ/dP\|_{P,2} := \int (dQ/dP)^2\, dP \leq B$ via Lemma 2, we get*

$$\mathbb{P}\left( P_e^{split}(\mathbf{D}_n) > \alpha + \frac{2B(2C+1)}{\delta\sqrt{m}} \right) \leq \delta, \qquad 0 < \delta \leq 1. \tag{8}$$

To assess the robustness of weighted conformal prediction methods, we compute training-conditional error bounds with a very mild assumption on $dQ/dP$. The proof of this theorem follows from Lemma 4 and the proof of Theorem 1.

**Theorem 2** *Assume that $Q$ is absolutely continuous with respect to $P$ ($Q \ll P$) and that $dQ/dP < \infty$ $Q$-a.s. Fix $B, \delta > 0$ and $\epsilon \in (0, 1/2)$. It holds that,*

$$P_e^{split}(\mathbf{D}_n) < \alpha + 3\delta + 4\,\mathbb{P}\left( \frac{dQ}{dP}(Z_{test}) > m \right) + 2C\sqrt{\frac{B}{m'}} + \mathbb{P}\left( \frac{dQ}{dP}(Z_{test}) > B \right),$$

*with probability at least,*

$$1 - 4e^{-m'\delta^2/(2B^2)} - 2e^{-2m^{2\epsilon}} - 2\,\mathbb{P}\left( \frac{dQ}{dP}(Z_{test}) > m \right)$$

$$- \frac{8}{m\delta^2}\mathbb{E}\left( \left( \frac{dQ}{dP}(Z_{calib}) \right)^2 \mathbf{1}\left\{ \frac{dQ}{dP}(Z_{calib}) \leq m \right\} \right),$$

*where $m' = m\,\mathbb{P}\left( \frac{dQ}{dP}(Z_{calib}) \leq B \right) - m^{1/2+\epsilon}$.*

### 3.2  Full conformal and jackknife+

Let $\mu_\beta : \mathcal{X} \to \mathbb{R}$ denote a predictor function parameterized by $\beta \in \mathbb{R}^p$. By a slight abuse of notation, let the map $T : \cup_{n \geq 1}(\mathcal{X} \times \mathcal{Y})^n \to \mathbb{R}^p$ denote a training algorithm for estimating $\beta$, hence, $\hat{\beta}_n = T(\mathbf{D}_n)$. In this case, we have $\hat{\mu}_{\mathbf{D}_n} = \mu_{\hat{\beta}_n}$.

**Assumption 1 (Uniform stability)** *For all $i \in [n]$, we have*

$$\sup_{z_1,\ldots,z_n} \|\mu_{T(z_1,\ldots,z_{i-1},z_{i+1}\ldots,z_n)} - \mu_{T(z_1,\ldots,z_i,\ldots,z_n)}\|_\infty \leq \frac{c_n}{2}.$$

In the case of the ridge regression Hoerl and Kennard (1970) with $\mathcal{Y} = [-I, I]$ and $\mathcal{X} = \{x : \|x\|_2 \leq b\}$, this assumption holds with $c_n = 16\, b^2 I^2/(\lambda\, n)$, where $\lambda$ denotes the regularization parameter. See Bousquet and Elisseeff (2002) for the general result on the uniform stability of RKHS regression.

**Assumption 2 (Bi-Lipschitz continuity)** *The map $\beta \mapsto \mu_\beta$ is bi-Lipschitz with respect to the $\infty$-norms, i.e.,*

$$\kappa_1 \|\beta - \beta'\|_\infty \ \leq \ \|\mu_\beta - \mu_{\beta'}\|_\infty \ \leq \ \kappa_2 \|\beta - \beta'\|_\infty,$$

*for some constants $0 < \kappa_1 \leq \kappa_2 < \infty$.*

**Remark 1** *It is worth noting that if the parameter space $\Theta \subseteq \mathbb{R}^p$ is compact, $\Phi : U \to L^\infty(\mathcal{X})$ given by $\beta \mapsto \mu_\beta$ is continuously differentiable for some open $U \supseteq \Theta$, then $\kappa_2 < \infty$. Moreover, the inverse function theorem (for Banach spaces), gives the sufficient condition under which the inverse is continuously differentiable over $\Phi(U)$ and hence $\kappa_1 > 0$.*

In the case of linear regression with $\mathcal{X} = \{x : \|x\|_2 \le b\}$, one can verify that Assumption 2 holds with $\kappa_1 = b$ and $\kappa_2 = \sqrt{p}b$. Let $\overline{\beta}_n = \mathbb{E}\hat{\beta}_n$, $\hat{\beta}_{-i} = T(Z_1, \ldots, Z_{i-1}, Z_{i+1}, \ldots, Z_n)$ with $Z_i = (X_i, Y_i)$ and $\overline{\beta}_{-i} = \mathbb{E}\hat{\beta}_{-i}$. Define

$$F^{(n-1)}(t) := \mathbb{P}_{Z_1 \sim P}\left(\left|Y_1 - \mu_{\overline{\beta}_{-1}}(X_1)\right| \le t\right),$$

$$F_Q^{(n-1)}(t) := \mathbb{P}_{Z_1 \sim Q}\left(\left|Y_1 - \mu_{\overline{\beta}_{-1}}(X_1)\right| \le t\right).$$

**Assumption 3 (Bounded density)** $F'^{(n)} < L_n$ and $F_Q'^{(n)} < L_{Q,n}$ where $F'^{(n)}$ and $F_Q'^{(n)}$ denote the derivative of $F^{(n)}$ and $F_Q^{(n)}$, respectively.

We introduce the following shorthand:

$$A(n, p, \epsilon) := 2\kappa_2 c_{n-1}\left(\frac{1}{\kappa_1} + \sqrt{\frac{n}{2\kappa_1^2}\log\frac{2p}{\epsilon}}\right).$$

**Theorem 3 (Jackknife+ under exchangeability)** *Define* $P_e^{J+}(\mathbf{D}_n) := \mathbb{P}\left(Y_{test} \notin \hat{C}_\alpha^{J+}(X_{test})\Big|\mathbf{D}_n\right)$. *Under Assumptions 1—3, for all $\epsilon, \delta > 0$, it holds that*

$$\mathbb{P}\left(P_e^{J+}(\mathbf{D}_n) > \alpha + \sqrt{\frac{\log(2/\delta)}{2n}} + L_{n-1} A(n, p, \epsilon)\right) \le \epsilon + \delta.$$

Now we present our result on jackknife+ under covariate shift when $dQ/dP \le B$.

**Theorem 4 (Jackknife+ under covariate shift)** *Assume that $Q$ is absolutely continuous with respect to $P$ ($Q \ll P$) and $dQ/dP \le B$. Under Assumptions 1—3, for all $\epsilon, \delta > 0$, it holds that*

$$\mathbb{P}\left(P_e^{J+}(\mathbf{D}_n) > \alpha + \left(\sqrt{2B\log 4/\delta} + 2C\right)\sqrt{\frac{B}{n}} + L_{Q,n-1} A(n, p, \epsilon)\right) \le \epsilon + \delta,$$

*where $C$ is a universal constant and $A(n, p, \epsilon)$ is the same as in Theorem 3.*

Using the same arguments as in the proof of this theorem, one can get a coverage bound for the CV+ as well. Unlike (2) which is meaningful only if the number of samples in each fold $\ell$ is large, the bound we present in the following corollary is suitable for cases where $\ell/n \to 0$.

**Corollary 2 (CV+)** *Define* $P_e^{CV+}(\mathbf{D}_n) := \mathbb{P}\left(Y_{test} \notin \hat{C}_\alpha^{CV+}(X_{test})\Big|\mathbf{D}_n\right)$. *Under Assumptions 1—3, for all $\epsilon, \delta > 0$, it holds that*

$$\mathbb{P}\left(P_e^{CV+}(\mathbf{D}_n) > \alpha + \sqrt{\frac{\log(2/\delta)}{2n}} + 2\ell L_{n-\ell} \kappa_2 c_{n-\ell}\left(\frac{1}{\kappa_1} + \sqrt{\frac{n}{2\kappa_1^2}\log\frac{2p}{\epsilon}}\right)\right) \le \epsilon + \delta.$$

The following theorem concerns the training-conditional guarantees for the full conformal prediction. We again introduce a shorthand:

$$E(n, p, \epsilon) := c_{n+1} + \sqrt{2n\log\frac{2p}{\epsilon}}\,\frac{\kappa_2 c_n}{\kappa_1}.$$

**Theorem 5 (Full conformal under exchangeability)** *Define* $P_e^{full}(\mathbf{D}_n) := \mathbb{P}\left(Y_{test} \notin \hat{C}_\alpha^{full}(X_{test})\Big|\mathbf{D}_n\right)$. *Under Assumptions 1—3, for all $\epsilon, \delta > 0$, it holds that*

$$\mathbb{P}\left(P_e^{full}(\mathbf{D}_n) > \alpha + \sqrt{\frac{\log(2/\delta)}{2n}} + L_n E(n, p, \epsilon)\right) \le \epsilon + \delta.$$

**Theorem 6 (Full conformal under covariate shift)** *Assume that $Q$ is absolutely continuous with respect to $P$ ($Q \ll P$) and $dQ/dP \leq B$. Under Assumptions 1—3, for all $\epsilon, \delta > 0$, it holds that*

$$\mathbb{P}\left( P_e^{full}(\mathbf{D}_n) > \alpha + \left( \sqrt{2B \log(4/\delta)} + 2C \right) \sqrt{\frac{B}{n}} + L_{Q,n} E(n, p, \epsilon) \right) \leq \epsilon + \delta.$$

*where $C$ is a universal constant and $E(n, p, \epsilon)$ is the same as in Theorem 5.*

We note that similar to Remark 1, one can relax assumption $dQ/dP \leq B$ to $\|dQ/dP\|_{P,2} \leq B$ and in this case the bounds for the jackknife+ and full conformal methods hold with the slow rate $O(1/(\delta\sqrt{n}))$ instead of $O(\log(1/\delta)/\sqrt{n})$.

## 4 Discussion

We have presented training-conditional coverage guarantees for weighted conformal prediction methods under covariate shift. The split conformal method has been analyzed for the black-box training algorithm while the full conformal and jackknife+ have been analyzed under three assumptions. Although Assumptions 1 and 2 have been verified only for the ridge regression in this paper, we conjecture that they are satisfied by a truncated version of the general RKHS regressions which we leave for future research. Truncated and sketched versions of the RKHS models have been extensively studied in the previous literature from the computational efficiency perspective (see Amini (2021); Williams and Seeger (2000); Zhang et al. (2013b); Alaoui and Mahoney (2015); Cortes et al. (2010) and references therein). The results in this paper quantify the training sample size (or calibration sample size for split conformal) needed for the coverage under covariate shift.

**Estimation of the likehood ratio.** A natural question here is how to include the error arising from estimating the likelihood ratios in the analysis. Define

$$\hat{w}_i := \frac{V_i}{V} := \frac{\frac{dQ}{dP}(Z_i)}{\sum_{i\in[n]} \frac{dQ}{dP}(Z_i)}, \qquad \breve{w}_i := \frac{\hat{V}_i}{\hat{V}} := \frac{\widehat{\frac{dQ}{dP}}(Z_i)}{\sum_{i\in[n]} \widehat{\frac{dQ}{dP}}(Z_i)}$$

and let $s(\cdot)$ denote a fixed score function. In this case, one needs to include the following term to the weighted DKW inequalities derived in this paper,

$$\sup_{t\in\mathbb{R}} \left| \sum_{i\in[n]} (\breve{w}_i - \hat{w}_i)\mathbf{1}\{s(Z_i) \leq t\} \right| \leq \sum_{i\in[n]} |\breve{w}_i - \hat{w}_i|$$

$$= \sum_{i\in[n]} \left| \frac{\hat{V}_i}{\hat{V}} - \frac{V_i}{V} \right|$$

$$\leq \frac{\sum_{i\in[n]} \left( \left| \hat{V}_i V - V_i V \right| + \left| V_i V - V_i \hat{V} \right| \right)}{V\hat{V}}$$

$$= \frac{V \sum_{i\in[n]} \left| \hat{V}_i - V_i \right| + \left| V - \hat{V} \right| \left( \sum_{i\in[n]} V_i \right)}{V\hat{V}}$$

$$= \frac{\sum_{i\in[n]} \left| \hat{V}_i - V_i \right| + \left| V - \hat{V} \right|}{\hat{V}} \leq 2\frac{\sum_{i\in[n]} \left| \hat{V}_i - V_i \right|}{\hat{V}},$$

where the concentration properties of this term around zero depends on the estimator of $dQ/dP(Z_i)$.

**Comparison with the $(m, n)$-stability.** The $(m, n)$-stability parameters were recently introduced in Liang and Barber (2023) and used to compute training-conditional coverage bounds for inflated full conformal and jackknife+ prediction intervals under exchangeability. Unlike uniform stability which is a distribution-free

property of a training process, $(m, n)$-stability depends on both the training algorithm and the distributions of the data as follows

$$\psi_{m,n}^{\text{out}} := \mathbb{E}\,|\hat{\mu}_{\mathbf{D}_n}(X_{\text{test}}) - \hat{\mu}_{\mathbf{D}_{n+m}}(X_{\text{test}})|, \tag{9}$$

$$\psi_{m,n}^{\text{in}} := \mathbb{E}\,|\hat{\mu}_{\mathbf{D}_n}(X_1) - \hat{\mu}_{\mathbf{D}_{n+m}}(X_1)|, \tag{10}$$

with $X_{\text{test}} \perp\!\!\!\perp \mathbf{D}_{n+m}$, $\mathbf{D}_{n+m} = ((X_1, Y_1), ..., (X_{n+m}, Y_{n+m}))$, and $\hat{\mu}_{\mathbf{D}_n}$ denotes the predictor function obtained by training on $\mathbf{D}_n$. Although weaker than uniform stability, these parameters are yet not well-understood in a practical sense. To elaborate on the difference between this approach and uniform stability, we evaluate their resulting training-conditional bounds for the ridge regression under exchangeability.

Assume $\mathcal{X} = \{x : \|x\| \le b\}$ and $\mathcal{Y} = [-I, I]$. As stated in the previous section, this regression model satisfies $c_n = 16\,b^2 I^2/(\lambda\,n)$, $\kappa_1 = b$ and $\kappa_2 = \sqrt{p}\,b$. Hence, we get the following bound for both full conformal and jackknife+ methods,

$$\mathbb{P}\left(P_e(\mathbf{D}_n) > \alpha + O\big(n^{-1/2}\big(\sqrt{\log(1/\delta)} + \sqrt{p\log(2p/\epsilon)}\big)\big)\right) \le \epsilon + \delta.$$

On the other hand, the following bound is proposed for the $\gamma$-inflated jackknife+ in Liang and Barber (2023),

$$\mathbb{P}\left(P_e^{\text{J+},\gamma}(\mathbf{D}_n) > \alpha + 3\sqrt{\frac{\log(1/\delta)}{\min(m, n)}} + 2\sqrt[3]{\frac{\psi_{m,n-1}^{\text{out}}}{\gamma}}\right) \le 3\delta + \sqrt[3]{\frac{\psi_{m,n-1}^{\text{out}}}{\gamma}}, \tag{11}$$

for all $m \ge 1$. We get $\psi_{m,n}^{\text{out}} = O(mc_n)$ since $\psi_{1,n}^{\text{out}} \le c_{n+1}/2$ by definition (9) and Assumption 1, and $\psi_{m,n}^{\text{out}} \le \sum_{k=n}^{n+m-1} \psi_{1,k}^{\text{out}}$ holds according to in Liang and Barber (2023, Lemma 5.2) . Substituting for $\psi_{m,n-1}^{\text{out}}$ in bound (11), we obtain

$$\mathbb{P}\left(P_e^{\text{J+},\gamma}(\mathbf{D}_n) > \alpha + O\left(\sqrt{\frac{\log(1/\delta)}{\min(m, n)}} + \sqrt[3]{\frac{m\,c_{n-1}}{\gamma}}\right)\right) \le 3\delta + O\left(\sqrt[3]{\frac{mc_{n-1}}{\gamma}}\right). \tag{12}$$

Letting $m^{-1/2} = (m/n)^{1/3}$ to balance the two terms $\sqrt{\frac{\log(1/\delta)}{\min(m,n)}}$ and $\sqrt[3]{mc_{n-1}/\gamma}$, we get $m = n^{2/5}$. By plugging $m = n^{2/5}$ into (12), we have

$$\mathbb{P}\left(P_e^{\text{J+},\gamma}(\mathbf{D}_n) > \alpha + O\big(n^{-1/5}\big(\sqrt{\log(1/\delta)} + \gamma^{-1/3}\big)\big)\right) \le 3\delta + O\left(n^{-1/5}\gamma^{-1/3}\right). \tag{13}$$

This bound, although dimension-free (i.e., does not depend on $p$), is very slow with respect to the sample size. In Liang and Barber (2023), the same bound as (11) is established for $\gamma$-inflated full conformal method except with $\psi_{m-1,n+1}^{\text{in}}$ instead of $\psi_{m,n-1}^{\text{out}}$. Hence, the same bound as (13) can be obtained for the $\gamma$-inflated full conformal method via $\psi_{m,n}^{\text{in}} = O(mc_n)$.

## 5  Conclusion

In this work, we have studied the training-conditional coverage bounds of full conformal, jackknife+, and CV+ prediction regions from a uniform stability perspective, which is well-understood for convexly regularized empirical risk minimization over reproducing kernel Hilbert spaces. We have derived new bounds via a concentration argument for the (estimated) predictor function. In the case of ridge regression, we have used the uniform stability parameter to derive a bound for $(m, n)$-stability and compare the resulting bounds from Liang and Barber (2023) to the bounds established in this paper. We have observed that our rates are faster in sample size but dependent on the dimension of the problem. Even though our work is theoretical in nature, it can potentially shed light on understanding a much broader downstream-task setups in reconstructive self-supervised learning (SSL), where the existing literature focuses on either linear regression or ridge regression (e.g., Lee et al. (2021); Teng et al. (2022); Du and Xiang (2024)). For split

conformal, our result allows for flexible downstream setups in SSL well beyond simple regressions, while for jackknife+ it would be interesting to reveal the interplay between uniform stability and excess risk analysis in SSL. Another worthwhile direction is from the robustness perspective in conformal prediction and detection problems, including adversarial and label noise settings (e.g., Gendler et al. (2021); Einbinder et al. (2024); Zhang et al. (2025)). For instance, one can study the robustness of training conditional guarantees under adversarial attacks during inference time.

## Acknowledgment

This work was supported in part by the National Science Foundation under Grant CCF-2611415.

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

# A Technical Lemmas: Weighted DKW Inequalities

## A.1 Background

In this section, we first briefly introduce the notion of *bracketing number* and four fundamental results, which we will be applying in our proofs.

**Bracketing Numbers.** A measure of complexity for a class of functions is the bracketing number. For a pair of functions $f$ and $g$, the bracket $[f, g]$ is defined as follows,

$$[f, g] = \{h : f \leq h \leq g\}.$$

A set of brackets $[f_1, g_1], \ldots, [f_n, g_n]$ is a $\epsilon$-$L^q(P)$ bracketing of function class $\mathcal{F}$ if $\mathcal{F} \subseteq \bigcup_{1 \leq i \leq n}[f_i, g_i]$ and $(\int |f_i - g_i|^q dP)^{1/q} \leq \epsilon$ for all $i \in [n]$. The bracketing number $N_{[]}(\epsilon, \mathcal{F}, L^q(P))$ is the size of the smallest $\epsilon$-$L^q(P)$ bracketing of $\mathcal{F}$.

**Theorem 7 (Van Der Vaart and Wellner (1996)(Theorem 2.14.2))** *Let $\mathcal{F}$ be a class of measurable functions with a measurable envelope $F$. Then, for some universal constant $C > 0$,*

$$\mathbb{E}\left(\sup_{f \in \mathcal{F}} \left| \sqrt{n} \int f \, d(P_n - P) \right| \right) \leq C \|F\|_{P,2} \int_0^1 \sqrt{1 + \log N_{[]}\left(\epsilon\|F\|_{P,2}, \mathcal{F}, L^2(P)\right)} \, d\epsilon. \tag{14}$$

**Theorem 8 (Lafferty et al. (2008)(Theorem 7.86))** *Let $\mathcal{F}$ be a class of measurable functions with $A = \sup_{f \in \mathcal{F}} \|f\|_{P,1}$ and $B = \sup_{f \in \mathcal{F}} \|f\|_\infty$. Then, for $\epsilon < 2A/3$,*

$$\mathbb{P}\left(\sup_{f \in \mathcal{F}} \left| \int f \, d(P_n - P) \right| > \epsilon \right) \leq 4 N_{[]}\left(\epsilon/8, \mathcal{F}, L^1(P)\right) \exp\left(-\frac{96 n \epsilon^2}{76 AB}\right). \tag{15}$$

**Theorem 9 (Durrett (2019) (Theorem 2.2.11))** *For each $n$ let $X_{n,k}$, $1 \leq k \leq n$ be independent. Let $b_n > 0$ with $b_n \to \infty$, and let $\overline{X}_{n,k} = X_{n,k}\mathbf{1}\{|X_{n,k}| \leq b_n\}$. Suppose that, as $n \to \infty$, (i) $\sum_{k=1}^n \mathbb{P}(|X_{n,k}| > b_n) \to 0$, and (ii) $b_n^{-2} \sum_{k=1}^n \mathbb{E}\,\overline{X}_{n,k}^2 \to 0$. If we let $S_n = X_{1,n} + \ldots + X_{n,n}$ and put $a_n = \sum_{k=1}^n \mathbb{E}\,\overline{X}_{n,k}$, then $(S_n - a_n)/b_n \to 0$ in probability.*

**Proof:** Fix $\varepsilon > 0$ and let $\overline{S}_n = \overline{X}_{1,n} + \ldots + \overline{X}_{n,n}$. We note

$$\mathbb{P}\left(\left|\frac{S_n - a_n}{b_n}\right| > \varepsilon\right) \leq \mathbb{P}(S_n \neq \overline{S}_n) + \mathbb{P}\left(\left|\frac{\overline{S}_n - a_n}{b_n}\right| > \varepsilon\right).$$

We bound the two terms separately. For the first term $\mathbb{P}(S_n \neq \overline{S}_n)$, observe that the event $\{S_n \neq \overline{S}_n\}$ occurs if and only if at least one summand is truncated, that is, $\{S_n \neq \overline{S}_n\} = \bigcup_{k=1}^n \{|X_{n,k}| > b_n\}$. Hence, by the union bound and assumption (i), $\mathbb{P}(S_n \neq \overline{S}_n) \leq \sum_{k=1}^n \mathbb{P}(|X_{n,k}| > b_n) \to 0$ as $n \to \infty$.

For the second term, $\mathbb{P}(|(\overline{S}_n - a_n)/b_n| > \varepsilon)$, note that $a_n = \mathbb{E}\,\overline{S}_n$. By Chebyshev's inequality,

$$\mathbb{P}\left(\left|\frac{\overline{S}_n - a_n}{b_n}\right| > \varepsilon\right) \leq \frac{1}{\varepsilon^2}\,\mathbb{E}\left[\left(\frac{\overline{S}_n - a_n}{b_n}\right)^2\right] = \frac{1}{\varepsilon^2 b_n^2}\,\mathrm{Var}(\overline{S}_n).$$

Because the variables $\overline{X}_{n,1}, \ldots, \overline{X}_{n,n}$ are independent and $\mathrm{Var}(Y) \leq \mathbb{E}Y^2$, we have $\mathrm{Var}(\overline{S}_n) \leq \sum_{k=1}^n \mathbb{E}\,\overline{X}_{n,k}^2$. Combining these inequalities with by assumption (ii), we have

$$\mathbb{P}\left(\left|\frac{\overline{S}_n - a_n}{b_n}\right| > \varepsilon\right) \leq \frac{1}{\varepsilon^2 b_n^2} \sum_{k=1}^n \mathbb{E}\,\overline{X}_{n,k}^2 \to 0, \text{ as } n \to \infty. \qquad \blacksquare$$

We can now apply this theorem to obtain the following result, also known as the Feller's weak law of large numbers without a first moment assumption.

**Theorem 10 (Durrett (2019) (Theorem 2.2.12))** *Let $X_1, X_2, \ldots$ be i.i.d. with $x\,\mathbb{P}(|X_i| > x) \to 0$ as $x \to \infty$. Let $S_n = X_1 + \ldots + X_n$ and let $\mu_n = \mathbb{E}\,X_1\mathbf{1}\{|X_1| \leq n\}$. Then $S_n/n - \mu_n \to 0$ in probability.*

**Proof:** We apply Durrett's Theorem 2.2.11 (i.e., Theorem 9) to the triangular array with entries $X_{n,k} = X_k$ for $1 \leq k \leq n$ and the normalizing constants $b_n = n$. It suffices to check the two conditions of that theorem.

**(i)** $\sum_{k=1}^n \mathbb{P}(|X_{n,k}| > n) = n\mathbb{P}(|X_1| > n) \to 0$ by the assumed tail condition $x\mathbb{P}(|X_1| > x) \to 0$.

**(ii)** Let $\overline{X}_{n,k} = X_k \mathbf{1}(|X_k| \leq n)$. We need to show $n^{-2} \sum_{k=1}^{n} \mathbb{E}(\overline{X}_{n,k}^2) = n^{-1} \mathbb{E}(\overline{X}_{n,1}^2) \longrightarrow 0$. By Lemma 2.2.13 of Durrett (2019) with $Y = |\overline{X}_{n,1}|$ and $p = 2$,

$$\mathbb{E}(\overline{X}_{n,1}^2) = \int_0^{\infty} 2y \, \mathbb{P}(|\overline{X}_{n,1}| > y) \, dy.$$

Since $|\overline{X}_{n,1}| \leq n$, the integrand vanishes for $y \geq n$, and for $0 \leq y \leq n$ we have

$$\mathbb{P}(|\overline{X}_{n,1}| > y) \leq \mathbb{P}(|X_1| > y).$$

Set $g(y) = 2y \, \mathbb{P}(|X_1| > y)$. Hence

$$\mathbb{E}(\overline{X}_{n,1}^2) \leq \int_0^n g(y) \, dy.$$

By assumption, we have $g(y) \to 0$, as $y \to \infty$, so does its average. ∎

## A.2 Weighted DKW Inequalities

With these results in place, we are ready to proceed with our proofs. Let $F_Q(x) = \mathbb{P}_{Z \sim Q}(s(Z) \leq x)$ where $s(\cdot)$ denotes some fixed score function. For $Z_i \overset{\text{i.i.d.}}{\sim} P$, we define

$$\hat{F}_{Q,n}(x) := \sum_{i \in [n]} \hat{w}_i \mathbf{1}\{s(Z_i) \leq x\}, \qquad \hat{w}_i = \frac{\frac{dQ}{dP}(Z_i)}{\sum_{i \in [n]} \frac{dQ}{dP}(Z_i)}. \tag{16}$$

**Lemma 1 (Bounded likelihood ratio)** *Assume that $Q$ is absolutely continuous with respect to $P$ ($Q \ll P$) and $dQ/dP \leq B$. Then, for all $\delta > 0$*

$$\mathbb{P}\left( \sup_{x \in \mathbb{R}} \left| \hat{F}_{Q,n}(x) - F_Q(x) \right| > \delta + 2C\sqrt{\frac{B}{n}} \right) \leq 4e^{-n\delta^2/(2B^2)},$$

*where $C > 0$ is some universal constant.*

**Proof:** By substituting the formula for $\hat{w}_i$ in the definition of $\hat{F}_{Q,n}$, we get

$$\hat{F}_{Q,n} = \frac{\frac{1}{n} \sum_{i \in [n]} \frac{dQ}{dP}(Z_i) \mathbf{1}\{s(Z_i) \leq x\}}{\frac{1}{n} \sum_{i \in [n]} \frac{dQ}{dP}(Z_i)} \ .$$

Hence we have

$$\sup_{x \in \mathbb{R}} |\hat{F}_{Q,n}(x) - F_Q(x)| \leq \sup_{x \in \mathbb{R}} \left| \frac{1}{n} \sum_{i \in [n]} \frac{dQ}{dP}(Z_i) \mathbf{1}\{s(Z_i) \leq x\} - F_Q(x) \right| +$$

$$\left| \frac{1}{\frac{1}{n} \sum_{i \in [n]} \frac{dQ}{dP}(Z_i)} - 1 \right| \sup_{x \in \mathbb{R}} \left| \frac{1}{n} \sum_{i \in [n]} \frac{dQ}{dP}(Z_i) \mathbf{1}\{s(Z_i) \leq x\} \right|$$

$$= \sup_{x \in \mathbb{R}} \left| \frac{1}{n} \sum_{i \in [n]} \frac{dQ}{dP}(Z_i) \mathbf{1}\{s(Z_i) \leq x\} - F_Q(x) \right| + \left| 1 - \frac{1}{n} \sum_{i \in [n]} \frac{dQ}{dP}(Z_i) \right|. \tag{17}$$

Regarding the first term, we have,

$$\sup_{x \in \mathbb{R}} \left| \frac{1}{n} \sum_{i \in [n]} \frac{dQ}{dP}(Z_i) \mathbf{1}\{s(Z_i) \leq x\} - F_Q(x) \right| = \sup_{x \in \mathbb{R}} \left| \int \frac{dQ}{dP}(z) \mathbf{1}\{s(z) \leq x\} \, d(P_n - P) \right|$$

$$= \sup_{x \in \mathbb{R}} \left| \int f_x \, d(P_n - P) \right|,$$

where $f_x \in L^1(\mathcal{X} \times \mathcal{Y}, P)$ is defined as $f_x(z) := \frac{dQ}{dP}(z)\mathbf{1}\{s(z) \le x\}$, and $P_n(A) = \frac{1}{n}\sum_{i \in [n]} \mathbf{1}\{Z_i \in A\}$ is the empirical measure.

We note,

$$1 \le \|dQ/dP\|_{P,2}^2 = \int (dQ/dP)^2 \, dP \le \|dQ/dP\|_\infty \int dQ/dP \, dP \le B.$$

according to the Holder's inequality. Hence, $\|dQ/dP\|_{P,2} \le \sqrt{B}$ and by Theorem 7 we have

$$\mathbb{E}\left( \sup_{x \in \mathbb{R}} \left| \int f_x \, d(P_n - P) \right| \right) \le C\sqrt{\frac{B}{n}} \int_0^1 \sqrt{1 + \log N_{[]}\left(\epsilon \|dQ/dP\|_{P,2}, \mathcal{F}, L^2(P)\right)} \, d\epsilon, \tag{18}$$

with some universal constant $C > 0$, and $N_{[]}\left(\epsilon \|dQ/dP\|_{P,2}, \mathcal{F}, L^2(P)\right)$ denotes the bracketting number for function class $\mathcal{F} = \{f_x : x \in \mathbb{R}\}$. It remains to compute an upper bound for $N_{[]}\left(\epsilon, \mathcal{F}, L^2(P)\right)$. Let $t_0 = -\infty$ and define,

$$t_{i+1} := \sup \left\{ t \in \mathbb{R} : \int f^2(z)\mathbf{1}\{s(z) \in (t_i, t]\} \, dP < \epsilon^2 \right\}, \tag{19}$$

where $f = dQ/dP$. We note that there exists $\ell < \infty$ such that $t_\ell = \infty$. This is true since $t_{j+1} < \infty$ implies

$$\int f^2(z)\mathbf{1}\{s(z) \in (t_i, t_{i+1}]\} \, dP \ge \epsilon^2, \quad 0 \le i \le j,$$

according to (19) and dominated convergence theorem. Hence,

$$(j+1)\epsilon^2 \le \sum_{i=0}^{j} \int f^2(z)\mathbf{1}\{s(z) \in (t_i, t_{i+1}]\} \, dP \le \int f^2 \, dP = \|dQ/dP\|_{P,2}^2, \tag{20}$$

and therefore $j + 1 \le (\|dQ/dP\|_{P,2}/\epsilon)^2$. Let $k = \min\{\ell \in \mathbb{Z} : t_\ell = \infty\}$. Now define $\epsilon$-brackets as follows

$$b_i = \{g(z) : f(z)\mathbf{1}\{s(z) \le t_i\} \le g(z) \le f(z)\mathbf{1}\{s(z) < t_{i+1}\}\}$$

where $0 \le i \le k - 1$. Clearly, $\mathcal{F} \subset \cup_{i=0}^{k} b_i$ with $b_k = \{f\}$. Brackets $b_0, \ldots, b_{k-1}$ have size

$$\left( \int f^2(z)\mathbf{1}\{s(z) \in (t_i, t_{i+1})\} \, dP \right)^{1/2} \le \epsilon, \quad 0 \le i \le k - 1,$$

according to (19) and bracket $b_k$ has size 0. Hence, we have $k + 1$ brackets in total with size smaller than $\epsilon$. By (20), we have $k - 1 \le (\|dQ/dP\|_{P,2}/\epsilon)^2$ which implies that $N_{[]}\left(\epsilon, \mathcal{F}, L^2(P)\right) \le k + 1 \le 2 + (\|dQ/dP\|_{P,2}/\epsilon)^2$. Plugging this into (20), we get

$$N_{[]}\left(\epsilon \|dQ/dP\|_{P,2}, \mathcal{F}, L^2(P)\right) \le 2 + \frac{1}{\epsilon^2}.$$

Therefore we have,

$$\mathbb{E}\left( \sup_{x \in \mathbb{R}} \left| \int f_x \, d(P_n - P) \right| \right) \le 2C\sqrt{\frac{B}{n}} = C'\sqrt{\frac{B}{n}}, \tag{21}$$

which follows from the observation that

$$\int_0^1 \sqrt{1 + \log(2 + 1/\epsilon^2)} \le \left( 1 + \int_0^1 \log(2 + 1/\epsilon^2) \right)^{1/2} \le \left( 1 + 2\int_0^1 (\epsilon^2 - \log \epsilon) \right)^{1/2} \le 2.$$

Now let $P_n^{(i)}$ denote the empirical measure obtained after replacing $Z_i$ by some arbitrary $Z_i'$. In this case, we observe that

$$\left| \sup_{x \in \mathbb{R}} \left| \int f_x \, d(P_n - P) \right| - \sup_{x \in \mathbb{R}} \left| \int f_x \, d(P_n^{(i)} - P) \right| \right| \le \sup_{x \in \mathbb{R}} \left| \int f_x \, d(P_n - P_n^{(i)}) \right|$$

$$\leq \frac{1}{n} \sup_{x \in \mathbb{R}} |f_x(Z_i) - f_x(Z_i')| \leq \frac{B}{n}.$$

Therefore, by McDiarmid's inequality McDiarmid et al. (1989) we get

$$\mathbb{P}\left( \left| \sup_{x \in \mathbb{R}} \left| \int f_x \ d(P_n - P) \right| - \mathbb{E}\left( \sup_{x \in \mathbb{R}} \left| \int f_x \ d(P_n - P) \right| \right) \right| > \delta \right) \leq 2e^{-2n\delta^2/B^2}.$$

Combining with (21), we obtain,

$$\mathbb{P}\left( \sup_{x \in \mathbb{R}} \left| \int f_x \ d(P_n - P) \right| > \delta + C'\sqrt{\frac{B}{n}} \right) \leq 2e^{-2n\delta^2/B^2}.$$

Back to (17), for the second term, Hoeffding's inequality implies

$$\mathbb{P}\left( \left| 1 - \frac{1}{n} \sum_{i \in [n]} \frac{dQ}{dP}(Z_i) \right| > \delta \right) \leq 2e^{-2n\delta^2/B^2}. \tag{22}$$

Combining the bounds for the two terms, we obtain,

$$\mathbb{P}\left( \sup_{x \in \mathbb{R}} \left| \hat{F}_{Q,n}(x) - F_Q(x) \right| > \delta + C'\sqrt{\frac{B}{n}} \right) \leq 4e^{-n\delta^2/(2B^2)}. \qquad \blacksquare$$

The following version relaxes the assumption $dQ/dP \leq B$ to $\|dQ/dP\|_{P,2} \leq K$ at the cost of slower rates.

**Lemma 2 (Alternative version: bounded second moment)** *Assume $Q \ll P$ and $\|dQ/dP\|_{P,2} \leq K$. Then, for all $\delta > 0$*

$$\mathbb{P}\left( \sup_{x \in \mathbb{R}} \left| \hat{F}_{Q,n}(x) - F_Q(x) \right| > \delta \right) \leq \frac{4\,CK}{\delta\sqrt{n}} + \frac{4(K^2 - 1)}{n\delta^2}.$$

*where $C > 0$ is some universal constant.*

**Proof:** By the same argument that led to (21) in the proof of Lemma 1, we get

$$\mathbb{E}\left( \sup_{x \in \mathbb{R}} \left| \int f_x \ d(P_n - P) \right| \right) \leq \frac{2CK}{\sqrt{n}} = \frac{C'K}{\sqrt{n}}.$$

Combining this with Markov's inequality, we obtain

$$\mathbb{P}\left( \sup_{x \in \mathbb{R}} \left| \int f_x \ d(P_n - P) \right| > \delta \right) \leq \frac{C'K}{\delta\sqrt{n}}. \tag{23}$$

This bounds the first term of (17). Regarding the second term, Chebyshev's inequality gives

$$\mathbb{P}\left( \left| 1 - \frac{1}{n} \sum_{i \in [n]} \frac{dQ}{dP}(Z_i) \right| > \delta \right) \leq \frac{\mathrm{Var}\left( \frac{1}{n} \sum_{i \in [n]} \frac{dQ}{dP}(Z_i) \right)}{\delta^2} \leq \frac{K^2 - 1}{n\delta^2}. \tag{24}$$

The result follows from combining (23) and (24). $\qquad \blacksquare$

In the following, we present yet another version by dropping the dependency on the constant $C$.

**Lemma 3 (Alternative version: bounded likelihood ratio)** *Assume $Q \ll P$ and $dQ/dP \leq B$. Then,*

$$\mathbb{P}\left( \sup_{x \in \mathbb{R}} |\hat{F}_{Q,n}(x) - F_Q(x)| > \delta \right) \leq (72/\delta)e^{-n\delta^2/(4B)} + 2e^{-n\delta^2/(2B^2)}$$

*for all $\delta > 0$.*

**Proof:** To prove this lemma, we compute the upper bound for the first term of (17) differently. According to Theorem 8, we get

$$\mathbb{P}\left(\sup_{x \in \mathbb{R}} \left| \int f_x \, d(P_n - P) \right| > \epsilon \right) \leq 4 N_{[]}(\epsilon/8, \mathcal{F}, L^1(P)) \, e^{-n\epsilon^2/B} \tag{25}$$

for $\epsilon \leq 2/3$, where $N_{[]}(\epsilon/8, \mathcal{F}, L^1(P))$ denotes the bracketting number for function class $\mathcal{F} = \{f_x : x \in \mathbb{R}\}$. Similar to the proof of Lemma 1, it can be shown that $N_{[]}(\epsilon, \mathcal{F}, L^1(P)) \leq 2 + 1/\epsilon$. Therefore,

$$\mathbb{P}\left(\sup_{x \in \mathbb{R}} \left| \int f_x \, d(P_n - P) \right| > \epsilon \right) \leq 8(1 + 4/\epsilon) \, e^{-n\epsilon^2/B} \leq (36/\epsilon) e^{-n\epsilon^2/B}.$$

Combining with (22) for the second term of (17), we obtain

$$\mathbb{P}\left(\sup_{x \in \mathbb{R}} |\hat{F}_{Q,n}(x) - F_Q(x)| > \delta \right) \leq (72/\delta) e^{-n\delta^2/(4B)} + 2e^{-n\delta^2/(2B^2)}$$

$$\leq (74/\delta) \exp\left(-\frac{n\delta^2}{2(B+1)^2}\right). \qquad \blacksquare$$

**Lemma 4 (Alternative version: unbounded likelihood ratio)** *Assume $Q \ll P$ and that $dQ/dP < \infty$ $Q$-a.s. Fix $B, \delta > 0$ and $\epsilon \in (0, 1/2)$. It holds that,*

$$\sup_{x \in \mathbb{R}} \left| F_Q(x) - \hat{F}_{Q,n}(x) \right| \leq 3\delta + 4 \, \mathbb{P}\left(\frac{dQ}{dP}(Z_{test}) > n\right) + C' \sqrt{\frac{B}{n'}} + \mathbb{P}\left(\frac{dQ}{dP}(Z_{test}) > B\right),$$

*with probability at least,*

$$1 - 4e^{-n'\delta^2/(2B^2)} - 2e^{-2n^{2\epsilon}} - 2 \, \mathbb{P}\left(\frac{dQ}{dP}(Z_{test}) > n\right)$$

$$- \frac{8}{n\delta^2} \mathbb{E}\left(\left(\frac{dQ}{dP}(Z_{calib})\right)^2 \mathbf{1}\left\{\frac{dQ}{dP}(Z_{calib}) \leq n\right\}\right),$$

*where $n' = n \, \mathbb{P}\left(\frac{dQ}{dP}(Z_{calib}) \leq B\right) - n^{1/2+\epsilon}$.*

**Proof:** To drop the assumption on $dQ/dP$, we start by writing $\mathcal{X} = \mathcal{X}_B \cup \mathcal{X}_B^c$, where $\mathcal{X}_B = \{x : dQ_X/dP_X(x) \leq B\}$. Define

$$F_{Q|\mathcal{X}_B} := \mathbb{P}\left(s(Z_{\text{test}}) \leq x \,\Big|\, \frac{dQ}{dP}(Z_{\text{test}}) \leq B\right),$$

$$\hat{F}_{Q,n|\mathcal{X}_B} := \frac{\sum_{i \in \mathcal{I}} \frac{dQ}{dP}(Z_i) \mathbf{1}\{s(Z_i) \leq x\} \mathbf{1}\left\{\frac{dQ}{dP}(Z_i) \leq B\right\}}{\sum_{i \in \mathcal{I}} \frac{dQ}{dP}(Z_i) \mathbf{1}\left\{\frac{dQ}{dP}(Z_i) \leq B\right\}},$$

where the summations are taken over calibration data points generated from $P$. We note,

$$\sup_{x \in \mathbb{R}} \left| F_Q(x) - \hat{F}_{Q,n}(x) \right| \leq \sup_{x \in \mathbb{R}} \left| F_Q(x) - F_{Q|\mathcal{X}_B}(x) \right| + \sup_{x \in \mathbb{R}} \left| F_{Q|\mathcal{X}_B}(x) - \hat{F}_{Q,n|\mathcal{X}_B}(x) \right|$$

$$+ \sup_{x \in \mathbb{R}} \left| \hat{F}_{Q,n|\mathcal{X}_B}(x) - \hat{F}_{Q,n}(x) \right|.$$

To upper-bound the first term, we observe that

$$\left| F_Q(x) - F_{Q|\mathcal{X}_B}(x) \right| = \left| \mathbb{P}\left(s(Z_{\text{test}}) \leq x\right) - \mathbb{P}\left(s(Z_{\text{test}}) \leq x \,\Big|\, \frac{dQ}{dP}(Z_{\text{test}}) \leq B\right) \right|$$

$$= \left| \mathbb{P}\left(s(Z_{\text{test}}) \leq x \,\Big|\, \frac{dQ}{dP}(Z_{\text{test}}) \leq B\right) \left(\mathbb{P}\left(\frac{dQ}{dP}(Z_{\text{test}}) \leq B\right) - 1\right) \right|$$

$$+ \mathbb{P}\left(s(Z_{\text{test}}) \le x \middle| \frac{dQ}{dP}(Z_{\text{test}}) > B\right) \mathbb{P}\left(\frac{dQ}{dP}(Z_{\text{test}}) > B\right)\Bigg|$$

$$\le \mathbb{P}\left(\frac{dQ}{dP}(Z_{\text{test}}) > B\right)\left|\mathbb{P}\left(s(Z_{\text{test}}) \le x \middle| \frac{dQ}{dP}(Z_{\text{test}}) > B\right)\right.$$

$$\left. - \mathbb{P}\left(s(Z_{\text{test}}) \le x \middle| \frac{dQ}{dP}(Z_{\text{test}}) \le B\right)\right|$$

$$\le \mathbb{P}\left(\frac{dQ}{dP}(Z_{\text{test}}) > B\right).$$

Regarding the second term, using the same argument as in the proof of Lemma 1, we get

$$\mathbb{P}\left(\sup_{x \in \mathbb{R}}\left|F_{Q|\mathcal{X}_B}(x) - \hat{F}_{Q,n|\mathcal{X}_B}(x)\right| \le \delta + C'\sqrt{B/n_B}\,\middle|\, n_B\right) \ge 1 - 4e^{-n_B\delta^2/(2B^2)},$$

where $n_B = \sum_{i \in \mathcal{I}} \mathbf{1}\left\{\frac{dQ}{dP}(Z_i) \le B\right\}$. Now let $n' = n\,\mathbb{P}\left(\frac{dQ}{dP}(Z_{\text{calib}}) \le B\right) - n^{1/2+\epsilon}$ and observe that

$$\mathbb{P}\left(\sup_{x \in \mathbb{R}}\left|F_{Q|\mathcal{X}_B}(x) - \hat{F}_{Q,n|\mathcal{X}_B}(x)\right| \le \delta + C'\sqrt{B/n'}\right)$$

$$\ge \mathbb{P}\left(\sup_{x \in \mathbb{R}}\left|F_{Q|\mathcal{X}_B}(x) - \hat{F}_{Q,n|\mathcal{X}_B}(x)\right| \le \delta + C'\sqrt{B/n_B},\, n_B \ge n'\right)$$

$$= \mathbb{E}\left(\mathbb{P}\left(\sup_{x \in \mathbb{R}}\left|F_{Q|\mathcal{X}_B}(x) - \hat{F}_{Q,n|\mathcal{X}_B}(x)\right| \le \delta + C'\sqrt{B/n_B}\middle| n_B\right)\mathbf{1}\{n_B \ge n'\}\right)$$

$$\ge \mathbb{E}\left(\left(1 - 4e^{-n_B\delta^2/(2B^2)}\right)\mathbf{1}\{n_B \ge n'\}\right)$$

$$\ge \left(1 - 4e^{-n'\delta^2/(2B^2)}\right)\mathbb{P}\left(n_B \ge n'\right)$$

$$\overset{(*)}{\ge} \left(1 - 4e^{-n'\delta^2/(2B^2)}\right)\left(1 - 2e^{-2n^{2\epsilon}}\right)$$

$$\ge 1 - 4e^{-n'\delta^2/(2B^2)} - 2e^{-2n^{2\epsilon}},$$

where $(*)$ holds according to Hoeffding's inequality.

For the third term, we note

$$|\hat{F}_{Q,n|\mathcal{X}_B}(x) - \hat{F}_{Q,n}(x)| = \left|\frac{W^<(x)}{W^<} - \frac{W^<(x) + W^>(x)}{W^< + W^>}\right|$$

$$= \left|\frac{W^<(x)W^> - W^<W^>(x)}{W^<(W^< + W^>)}\right|$$

$$\le 2\,\frac{W^<W^>}{W^<(W^< + W^>)} = 2\,\frac{W^>}{W^< + W^>}, \tag{26}$$

where

$$W^<(x) = \sum_{i \in \mathcal{I}} \frac{dQ}{dP}(Z_i)\mathbf{1}\{s(Z_i) \le x\}\mathbf{1}\left\{\frac{dQ}{dP}(Z_i) \le B\right\},$$

$$W^>(x) = \sum_{i \in \mathcal{I}} \frac{dQ}{dP}(Z_i)\mathbf{1}\{s(Z_i) \le x\}\mathbf{1}\left\{\frac{dQ}{dP}(Z_i) > B\right\},$$

$$W^< = \sum_{i \in \mathcal{I}} \frac{dQ}{dP}(Z_i)\mathbf{1}\left\{\frac{dQ}{dP}(Z_i) \le B\right\},$$

$$W^> = \sum_{i \in \mathcal{I}} \frac{dQ}{dP}(Z_i)\mathbf{1}\left\{\frac{dQ}{dP}(Z_i) > B\right\},$$

and the inequalty 26 follows from the fact that $W^<(x) \leq W^<$ and $W^>(x) \leq W^>$ for all $x \in \mathbb{R}$. Hence,

$$\sup_{x \in \mathbb{R}} |\hat{F}_{Q,n|\mathcal{X}_B}(x) - \hat{F}_{Q,n}(x)| \leq 2 \frac{\sum_{i \in \mathcal{I}} \frac{dQ}{dP}(Z_i) \mathbf{1}\left\{\frac{dQ}{dP}(Z_i) > B\right\}}{\sum_{i \in \mathcal{I}} \frac{dQ}{dP}(Z_i)} \xrightarrow{a.s.} 2 \, \mathbb{P}\left(\frac{dQ}{dP}(Z_{\text{test}}) > B\right).$$

We note,

$$\left| \frac{\frac{1}{n} \sum_{i \in \mathcal{I}} \frac{dQ}{dP}(Z_i) \mathbf{1}\left\{\frac{dQ}{dP}(Z_i) > B\right\}}{\frac{1}{n} \sum_{i \in \mathcal{I}} \frac{dQ}{dP}(Z_i)} - \mathbb{P}\left(\frac{dQ}{dP}(Z_{\text{test}}) > B\right) \right| \leq$$

$$\left| \frac{1}{n} \sum_{i \in \mathcal{I}} \frac{dQ}{dP}(Z_i) \mathbf{1}\left\{\frac{dQ}{dP}(Z_i) > B\right\} - \mathbb{P}\left(\frac{dQ}{dP}(Z_{\text{test}}) > B\right) \right| +$$

$$\left| \frac{1}{\frac{1}{n} \sum_{i \in \mathcal{I}} \frac{dQ}{dP}(Z_i)} - 1 \right| \left| \frac{1}{n} \sum_{i \in \mathcal{I}} \frac{dQ}{dP}(Z_i) \mathbf{1}\left\{\frac{dQ}{dP}(Z_i) > B\right\} \right| \leq$$

$$\left| \frac{1}{n} \sum_{i \in \mathcal{I}} \frac{dQ}{dP}(Z_i) \mathbf{1}\left\{\frac{dQ}{dP}(Z_i) > B\right\} - \mathbb{P}\left(\frac{dQ}{dP}(Z_{\text{test}}) > B\right) \right| + \left| \frac{1}{n} \sum_{i \in \mathcal{I}} \frac{dQ}{dP}(Z_i) - 1 \right| \leq$$

$$\left| \frac{1}{n} \sum_{i \in \mathcal{I}} \frac{dQ}{dP}(Z_i) \mathbf{1}\left\{\frac{dQ}{dP}(Z_i) > B\right\} - \mathbb{P}\left(B < \frac{dQ}{dP}(Z_{\text{test}}) \leq n\right) \right| +$$

$$\mathbb{P}\left(\frac{dQ}{dP}(Z_{\text{test}}) > B, \frac{dQ}{dP}(Z_{\text{test}}) > n\right) + \left| \frac{1}{n} \sum_{i \in \mathcal{I}} \frac{dQ}{dP}(Z_i) - \mathbb{P}\left(\frac{dQ}{dP}(Z_{\text{test}}) \leq n\right) \right| + \mathbb{P}\left(\frac{dQ}{dP}(Z_{\text{test}}) > n\right)$$

$$\leq \delta + 2 \, \mathbb{P}\left(\frac{dQ}{dP}(Z_{\text{test}}) > n\right),$$

with probability at least

$$1 - 2 \, \mathbb{P}\left(\frac{dQ}{dP}(Z_{\text{test}}) > n\right) - \frac{8}{n\delta^2} \mathbb{E}\left(\left(\frac{dQ}{dP}(Z_{\text{calib}})\right)^2 \mathbf{1}\left\{\frac{dQ}{dP}(Z_{\text{calib}}) \leq n\right\}\right), \tag{27}$$

where in the last step we have used (the proof of) Theorem 9 with $\epsilon = \delta/2$ and $b_n = n$ twice; once for the first term with $X_{n,k} = \frac{dQ}{dP}(Z_k) \mathbf{1}\left\{\frac{dQ}{dP}(Z_k) > B\right\}$, $k \in \mathcal{I}$, and the other time for the third term with $X_{n,k} = dQ/dP(Z_k)$, $k \in \mathcal{I}$. The requirements of the theorem are satisfied by

$$x \, \mathbb{P}\left(\frac{dQ}{dP}(Z_{\text{calib}}) > x\right) \leq \mathbb{E}\left(\frac{dQ}{dP}(Z_{\text{calib}}) \mathbf{1}\left\{\frac{dQ}{dP}(Z_{\text{calib}}) > x\right\}\right) = \mathbb{P}\left(\frac{dQ}{dP}(Z_{\text{test}}) > x\right) \to 0 \tag{28}$$

as $x \to \infty$ and then according to Theorem 10 ; also recall that $\mathcal{I} \subset [n]$ for the split conformal method corresponds to the calibration dataset. Putting everything together, we have

$$\sup_{x \in \mathbb{R}} \left| F_Q(x) - \hat{F}_{Q,n}(x) \right| \quad \leq \quad 3\delta \quad + \quad 4 \, \mathbb{P}\left(\frac{dQ}{dP}(Z_{\text{test}}) > n\right) \quad + \quad C'\sqrt{\frac{B}{n'}} \quad + \quad \mathbb{P}\left(\frac{dQ}{dP}(Z_{\text{test}}) > B\right),$$

with probability at least,

$$1 - 4e^{-n'\delta^2/(2B^2)} - 2e^{-2n^{2\epsilon}} - 2 \, \mathbb{P}\left(\frac{dQ}{dP}(Z_{\text{test}}) > n\right)$$

$$- \frac{8}{n\delta^2} \mathbb{E}\left(\left(\frac{dQ}{dP}(Z_{\text{calib}})\right)^2 \mathbf{1}\left\{\frac{dQ}{dP}(Z_{\text{calib}}) \leq n\right\}\right).$$

# B   Proof of Theorem 1

Recall from Section 2.3, we have

$$F_{Q,|\mathcal{I}|}(x) = w_{\text{test}}\mathbf{1}\{t = \infty\} + \sum_{i \in \mathcal{I}} w_i \mathbf{1}\{s(Z_i) \leq x\}, \qquad w_i = \frac{\frac{dQ}{dP}(Z_i)}{\frac{dQ}{dP}(Z_{\text{test}}) + \sum_{i \in \mathcal{I}} \frac{dQ}{dP}(Z_i)}, \qquad (29)$$

where $Z_i \overset{\text{i.i.d.}}{\sim} P$, $Z_{\text{test}} \sim Q$, and $\mathcal{I}$ denotes the set of indices corresponding to the calibration dataset. We note

$$
\begin{aligned}
P_e(\mathbf{D}_n) = \mathbb{P}\left(Y_{\text{test}} \notin \hat{C}_\alpha^{\text{split}}(X_{\text{test}}) \Big| \mathbf{D}_n\right) &= \mathbb{P}\left(s(X_{\text{test}}, Y_{\text{test}}; \hat{\mu}) > F_{Q,m}^{-1}(1-\alpha) \Big| \mathbf{D}_n\right) \\
&\leq \mathbb{P}\left(s(X_{\text{test}}, Y_{\text{test}}; \hat{\mu}) > \hat{F}_{Q,m}^{-1}(1-\alpha) \Big| \mathbf{D}_n\right) \\
&= 1 - F_Q(\hat{F}_{Q,m}^{-1}(1-\alpha)),
\end{aligned}
$$

where $F_Q(x) = \mathbb{P}_{Z \sim Q}(s(Z) \leq x)$ and $\hat{F}_{Q,m}^{-1}$ is defined according to (16) with index set $\mathcal{I}$ instead of $[n]$. Using the weighted DKW inequality from Lemma 1 we get

$$F_Q(\hat{F}_{Q,m}^{-1}(1-\alpha)) \geq \hat{F}_{Q,m}(\hat{F}_{Q,m}^{-1}(1-\alpha)) - \delta - C'\sqrt{\frac{B}{m}} \geq 1 - \alpha - \delta - C'\sqrt{\frac{B}{m}}, \quad \text{whp.}$$

Hence, we have

$$\mathbb{P}\left(P_e(\mathbf{D}_n) > \alpha + \delta + C'\sqrt{\frac{B}{m}}\right) \leq 4e^{-m\delta^2/(2B^2)}$$

for all $\delta > 0$, or equivalently,

$$\mathbb{P}\left(P_e(\mathbf{D}_n) > \alpha + \sqrt{\frac{2B^2}{m}\log\frac{4}{\delta}} + C'\sqrt{\frac{B}{m}}\right) = \mathbb{P}\left(P_e(\mathbf{D}_n) > \alpha + \left(\sqrt{2B\log 4/\delta} + C'\right)\sqrt{\frac{B}{m}}\right)$$
$$\leq \delta.$$

This completes the proof of Theorem 1.

## B.1   Proof of Remark 1

To see (8), we note that by using Lemma 2 instead of Lemma 1 we get

$$\mathbb{P}\left(P_e(\mathbf{D}_n) > \alpha + \delta\right) \leq \frac{6\,CK}{\delta\sqrt{m}} + \frac{4(K^2 - 1)}{m\delta^2}$$

for all $\delta > 0$. Hence,

$$\mathbb{P}\left(P_e(\mathbf{D}_n) > \alpha + 2\left(\frac{2CK + \sqrt{\delta(K^2 - 1)}}{\delta\sqrt{m}}\right)\right) \leq \delta$$

for all $\delta > 0$, which implies

$$\mathbb{P}\left(P_e(\mathbf{D}_n) > \alpha + \frac{2K(2C + 1)}{\delta\sqrt{m}}\right) \leq \delta, \qquad 0 < \delta \leq 1.$$

■.

# C    Proof for Jackknife+

**Lemma 5** *If Assumption 1 and 2 hold, then*

$$\mathbb{P}\left(\left\|\hat{\beta}_n - \mathbb{E}\,\hat{\beta}_n\right\|_\infty \geq \epsilon\right) \leq 2p\exp\left(-\frac{2\kappa_1^2\epsilon^2}{nc_n^2}\right).$$

**Proof:** Assumption 1 and 2 imply that

$$\sup_{z_1,\ldots,z_n,z_i'}\|T(z_1,\ldots,z_i\ldots,z_n) - T(z_1,\ldots,z_i',\ldots,z_n)\|_\infty \leq \frac{c_n}{\kappa_1}.$$

By McDiarmid's inequality McDiarmid et al. (1989) we get

$$\mathbb{P}\left(\|\hat{\beta}_n - \mathbb{E}\,\hat{\beta}_n\|_\infty \geq \epsilon\right) = \mathbb{P}\left(\left\|T(Z_1,\ldots,Z_n) - \mathbb{E}\,T(Z_1,\ldots,Z_n)\right\|_\infty \geq \epsilon\right) \leq 2p\exp\left(-\frac{2\kappa_1^2\epsilon^2}{nc_n^2}\right)$$

for independent $Z_i$ and all $\epsilon > 0$. ∎

**Lemma 6** *Under Assumptions 1 and 2 we have*

$$\mathbb{P}\left(\max_i \left\|\mu_{\hat{\beta}_{-i}} - \mu_{\overline{\beta}_{-1}}\right\|_\infty \geq \epsilon\right) \leq 2p\exp\left(-\frac{2\kappa_1^2}{n}\left(\frac{\epsilon}{\kappa_2 c_{n-1}} - \frac{1}{\kappa_1}\right)^2\right).$$

**Proof:** From Assumption 1 and 2, it follows that

$$\max_{i,j}\|\hat{\beta}_{-i} - \hat{\beta}_{-j}\|_\infty \leq \frac{c_{n-1}}{\kappa_1}. \tag{30}$$

Also, according to (5), we have $\|\hat{\beta}_{-1} - \overline{\beta}_{-1}\|_\infty < \epsilon$ with probability at least $1 - 2p\exp(-2\kappa_1^2\epsilon^2/(nc_{n-1}^2))$. We note that,

$$\mathbb{P}\left(\max_i\left\|\mu_{\hat{\beta}_{-i}} - \mu_{\overline{\beta}_{-1}}\right\|_\infty \geq \epsilon\right) \overset{(*)}{\leq} \mathbb{P}\left(\kappa_2\max_i\left\|\hat{\beta}_{-i} - \overline{\beta}_{-1}\right\|_\infty \geq \epsilon\right)$$

$$\overset{(**)}{\leq} \mathbb{P}\left(\kappa_2\left(\frac{c_{n-1}}{\kappa_1} + \left\|\hat{\beta}_{-1} - \overline{\beta}_{-1}\right\|_\infty\right) \geq \epsilon\right)$$

$$\leq 2p\exp\left(-\frac{2\kappa_1^2}{n}\left(\frac{\epsilon}{\kappa_2 c_{n-1}} - \frac{1}{\kappa_1}\right)^2\right),$$

where (*) and (**) hold according to Assumption 2 and (30), respectively. ∎

## C.1    Proof of Theorem 3

We note,

$$\hat{\mathcal{C}}_\alpha^{\mathrm{J+}}(X_{\text{test}}) \supseteq \left\{y \in \mathbb{R} : \frac{1}{n}\sum_{i=1}^n \mathbf{1}\left\{\left|Y_i - \mu_{\hat{\beta}_{-i}}(X_i)\right| \geq \left|y - \mu_{\hat{\beta}_{-i}}(X_{\text{test}})\right|\right\} > \alpha\right\}$$

$$\supseteq \left\{y \in \mathbb{R} : \frac{1}{n}\sum_{i=1}^n \mathbf{1}\left\{\left|Y_i - \mu_{\overline{\beta}_{-1}}(X_i)\right| - \left|\mu_{\hat{\beta}_{-i}}(X_i) - \mu_{\overline{\beta}_{-1}}(X_i)\right| \geq\right.\right.$$

$$\left.\left.\left|y - \mu_{\overline{\beta}_{-1}}(X_{\text{test}})\right| + \left|\mu_{\hat{\beta}_{-i}}(X_{\text{test}}) - \mu_{\overline{\beta}_{-1}}(X_{\text{test}})\right|\right\} > \alpha\right\},$$

where the first relation holds according to Bian and Barber (2023). This first step can also be recovered by letting $w_i = 1/(n+1)$ and $\hat{w}_i = 1/n$ in the first four steps of C.2. Assuming $\max_i \|\mu_{\hat{\beta}_{-i}} - \mu_{\overline{\beta}_{-1}}\|_\infty < \epsilon$, we obtain

$$
\hat{\mathcal{C}}_\alpha^{\text{J+}}(X_{\text{test}}) \supseteq \left\{ y \in \mathbb{R} : \frac{1}{n} \sum_{i=1}^n \mathbf{1}\left\{ \left|Y_i - \mu_{\overline{\beta}_{-1}}(X_i)\right| \geq \left|y - \mu_{\overline{\beta}_{-1}}(X_{\text{test}})\right| + 2\epsilon \right\} > \alpha \right\}
$$

$$
\supseteq \left\{ y \in \mathbb{R} : 1 - \hat{F}^{(n-1)}\left( \left|y - \mu_{\overline{\beta}_{-1}}(X_{\text{test}})\right| + 2\epsilon \right) > \alpha \right\}.
$$

Assuming $\left\|\hat{F}^{(n-1)} - F^{(n-1)}\right\|_\infty < \delta$, we obtain

$$
\hat{\mathcal{C}}_\alpha^{\text{J+}}(X_{\text{test}}) \supseteq \left\{ y \in \mathbb{R} : 1 - F^{(n-1)}\left( \left|y - \mu_{\overline{\beta}_{-1}}(X_{\text{test}})\right| + 2\epsilon \right) > \alpha + \delta \right\}
$$

$$
\supseteq \left\{ y \in \mathbb{R} : 1 - F^{(n-1)}\left( \left|y - \mu_{\overline{\beta}_{-1}}(X_{\text{test}})\right| \right) > \alpha + \delta + 2\epsilon L_{n-1} \right\}.
$$

Therefore,

$$
P_e(\mathbf{D}_n) = \mathbb{P}(Y_{\text{test}} \notin \hat{\mathcal{C}}_\alpha^{\text{J+}}(X_{\text{test}})|\mathbf{D}_n) \leq \mathbb{P}\left( 1 - F^{(n-1)}\left( \left|Y_{\text{test}} - \mu_{\overline{\beta}_{-1}}(X_{\text{test}})\right| \right) \leq \alpha + \delta + 2\epsilon L_{n-1} \right)
$$

$$
= \alpha + \delta + 2\epsilon L_{n-1}
$$

for $\mathbf{D}_n \in \mathcal{A} \cap \mathcal{B}$ where $\mathcal{A} := \left\{ D : \max_i \|\mu_{\hat{\beta}_{-i}} - \mu_{\overline{\beta}_{-1}}\|_\infty < \epsilon \right\}$ and $\mathcal{B} := \left\{ D : \left\|\hat{F}^{(n-1)} - F^{(n-1)}\right\|_\infty < \delta \right\}$. From Lemma 6, we know

$$
\mathbb{P}(\mathbf{D}_n \notin \mathcal{A}) \leq 2p \exp\left( -\frac{2\kappa_1^2}{n}\left( \frac{\epsilon}{\kappa_2 c_{n-1}} - \frac{1}{\kappa_1} \right)^2 \right).
$$

Also, according to Dvoretzky–Kiefer–Wolfowitz inequality Dvoretzky et al. (1956), we have $\mathbb{P}(\mathbf{D}_n \notin \mathcal{B}) \leq 2e^{-2n\delta^2}$. Thus,

$$
\mathbb{P}(P_e(\mathbf{D}_n) > \alpha + \delta + \epsilon) \leq \mathbb{P}((\mathcal{A} \cap \mathcal{B})^c) \leq 2e^{-2n\delta^2} + 2p \exp\left( -\frac{2\kappa_1^2}{n}\left( \frac{\epsilon}{2L_{n-1}\kappa_2 c_{n-1}} - \frac{1}{\kappa_1} \right)^2 \right),
$$

or equivalently,

$$
\mathbb{P}\left( P_e(\mathbf{D}_n) > \alpha + \sqrt{\frac{\log(2/\delta)}{2n}} + 2L_{n-1}\kappa_2 c_{n-1}\left( \frac{1}{\kappa_1} + \sqrt{\frac{n}{2\kappa_1^2}\log\frac{2p}{\epsilon}} \right) \right) \leq \epsilon + \delta. \quad \blacksquare
$$

## C.2 Proof of Theorem 4

We note,

$$
\hat{\mathcal{C}}_\alpha^{\text{J+}}(X_{\text{test}}) = \left\{ y \in \mathbb{R} : \sum_{i=1}^n w_i \mathbf{1}\left\{ \mu_{\hat{\beta}_{-i}}(X_{\text{test}}) + \left|Y_i - \mu_{\hat{\beta}_{-i}}(X_i)\right| < y \right\} < 1 - \alpha \right\} \bigcap
$$

$$
\left\{ y \in \mathbb{R} : \sum_{i=1}^n w_i \mathbf{1}\left\{ \mu_{\hat{\beta}_{-i}}(X_{\text{test}}) - \left|Y_i - \mu_{\hat{\beta}_{-i}}(X_i)\right| > y \right\} \leq 1 - \alpha \right\}
$$

$$
\supseteq \left\{ y \in \mathbb{R} : \sum_{i=1}^n w_i \mathbf{1}\left\{ \left|Y_i - \mu_{\hat{\beta}_{-i}}(X_i)\right| < \left|y - \mu_{\hat{\beta}_{-i}}(X_{\text{test}})\right| \right\} < 1 - \alpha \right\} \bigcap
$$

$$\left\{y \in \mathbb{R} : \sum_{i=1}^{n} w_i \mathbf{1}\left\{\left|\mu_{\hat{\beta}_{-i}}(X_{\text{test}}) - y\right| > \left|Y_i - \mu_{\hat{\beta}_{-i}}(X_i)\right|\right\} < 1 - \alpha\right\}$$

$$= \left\{y \in \mathbb{R} : \sum_{i=1}^{n} w_i \mathbf{1}\left\{\left|Y_i - \mu_{\hat{\beta}_{-i}}(X_i)\right| < \left|y - \mu_{\hat{\beta}_{-i}}(X_{\text{test}})\right|\right\} < 1 - \alpha\right\}$$

$$\supseteq \left\{y \in \mathbb{R} : \sum_{i=1}^{n} \hat{w}_i \mathbf{1}\left\{\left|Y_i - \mu_{\hat{\beta}_{-i}}(X_i)\right| < \left|y - \mu_{\hat{\beta}_{-i}}(X_{\text{test}})\right|\right\} < 1 - \alpha\right\}$$

$$= \left\{y \in \mathbb{R} : \sum_{i=1}^{n} \hat{w}_i \mathbf{1}\left\{\left|Y_i - \mu_{\hat{\beta}_{-i}}(X_i)\right| \geq \left|y - \mu_{\hat{\beta}_{-i}}(X_{\text{test}})\right|\right\} > \alpha\right\}$$

$$\supseteq \left\{y \in \mathbb{R} : \sum_{i=1}^{n} \hat{w}_i \mathbf{1}\left\{\left|Y_i - \mu_{\overline{\beta}_{-1}}(X_i)\right| - \left|\mu_{\hat{\beta}_{-i}}(X_i) - \mu_{\overline{\beta}_{-1}}(X_i)\right| \geq \right.\right.$$
$$\left.\left. \left|y - \mu_{\overline{\beta}_{-1}}(X_{\text{test}})\right| + \left|\mu_{\hat{\beta}_{-i}}(X_{\text{test}}) - \mu_{\overline{\beta}_{-1}}(X_{\text{test}})\right|\right\} > \alpha\right\},$$

where the first and last relations hold by the definition of $\hat{\mathcal{C}}_\alpha^{\text{J+}}(X_{\text{test}})$ and triangle inequality, respectively. Assuming $\max_i \|\mu_{\hat{\beta}_{-i}} - \mu_{\overline{\beta}_{-1}}\|_\infty < \epsilon$, we obtain

$$\hat{\mathcal{C}}_\alpha^{\text{J+}}(X_{\text{test}}) \supseteq \left\{y \in \mathbb{R} : \sum_{i=1}^{n} \hat{w}_i \mathbf{1}\left\{\left|Y_i - \mu_{\overline{\beta}_{-1}}(X_i)\right| \geq \left|y - \mu_{\overline{\beta}_{-1}}(X_{\text{test}})\right| + 2\epsilon\right\} > \alpha\right\}$$

$$\supseteq \left\{y \in \mathbb{R} : \sum_{i=1}^{n} \hat{w}_i \mathbf{1}\left\{\left|Y_i - \mu_{\overline{\beta}_{-1}}(X_i)\right| \geq \left|y - \mu_{\overline{\beta}_{-1}}(X_{\text{test}})\right| + 2\epsilon\right\} > \alpha\right\}$$

$$\supseteq \left\{y \in \mathbb{R} : 1 - \hat{F}_Q^{(n-1)}\left(\left|y - \mu_{\overline{\beta}_{-1}}(X_{\text{test}})\right| + 2\epsilon\right) > \alpha\right\},$$

where $\hat{F}_Q^{(n-1)}(t) := \sum_{i=1}^{n} \hat{w}_i \mathbf{1}\left\{\left|Y_i - \mu_{\overline{\beta}_{-1}}(X_i)\right| \leq t\right\}$ and, $\hat{w}_i$ and $w_i$ are defined in (16) and (29), respectively. Define,

$$F_Q^{(n-1)}(t) := \mathbb{P}_{Z_1 \sim Q}\left(\left|Y_1 - \mu_{\overline{\beta}_{-1}}(X_1)\right| \leq t\right).$$

Assuming $\left\|\hat{F}_Q^{(n-1)} - F_Q^{(n-1)}\right\|_\infty < \delta$, we obtain

$$\hat{\mathcal{C}}_\alpha^{\text{J+}}(X_{\text{test}}) \supseteq \left\{y \in \mathbb{R} : 1 - F_Q^{(n-1)}\left(\left|y - \mu_{\overline{\beta}_{-1}}(X_{\text{test}})\right| + 2\epsilon\right) > \alpha + \delta\right\}$$

$$\supseteq \left\{y \in \mathbb{R} : 1 - F_Q^{(n-1)}\left(\left|y - \mu_{\overline{\beta}_{-1}}(X_{\text{test}})\right|\right) > \alpha + \delta + 2\epsilon L_{Q,n-1}\right\}.$$

Therefore,

$$P_e(\mathbf{D}_n) = \mathbb{P}(Y_{\text{test}} \notin \hat{\mathcal{C}}_\alpha^{\text{J+}}(X_{\text{test}})|\mathbf{D}_n) \leq \mathbb{P}\left(1 - F_Q^{(n-1)}\left(\left|Y_{\text{test}} - \mu_{\overline{\beta}_{-1}}(X_{\text{test}})\right|\right) \leq \alpha + \delta + 2\epsilon L_{Q,n-1}\right)$$
$$= \alpha + \delta + 2\epsilon L_{Q,n-1}$$

for $\mathbf{D}_n \in \mathcal{A} \cap \mathcal{B}(\delta)$ where $\mathcal{A} := \left\{D : \max_i \|\mu_{\hat{\beta}_{-i}} - \mu_{\overline{\beta}_{-1}}\|_\infty < \epsilon\right\}$ and $\mathcal{B}(\delta) := \left\{D : \left\|\hat{F}_Q^{(n-1)} - F_Q^{(n-1)}\right\|_\infty \leq \delta\right\}$. From Lemma 6, we know

$$\mathbb{P}(\mathbf{D}_n \notin \mathcal{A}) \leq 2p \exp\left(-\frac{2\kappa_1^2}{n}\left(\frac{\epsilon}{\kappa_2 c_{n-1}} - \frac{1}{\kappa_1}\right)^2\right).$$

Also, according to weighted DKW inequality from Lemma 1, we have

$$\mathbb{P}\left(\mathbf{D}_n \notin \mathcal{B}\left(\delta + 2C\sqrt{\frac{B}{n}}\right)\right) \le 4e^{-n\delta^2/(2B^2)}.$$

Thus,

$$\mathbb{P}\left(P_e(\mathbf{D}_n) > \alpha + \delta + \epsilon + 2C\sqrt{\frac{B}{n}}\right) \le 4e^{-n\delta^2/(2B^2)}$$
$$+ 2p\exp\left(-\frac{2\kappa_1^2}{n}\left(\frac{\epsilon}{2L_{Q,n-1}\kappa_2 c_{n-1}} - \frac{1}{\kappa_1}\right)^2\right),$$

or equivalently,

$$\mathbb{P}\left(P_e(\mathbf{D}_n) > \alpha + \left(\sqrt{2B\log\frac{4}{\delta}} + 2C\right)\sqrt{\frac{B}{n}} + 2L_{Q,n-1}\kappa_2 c_{n-1}\left(\frac{1}{\kappa_1} + \sqrt{\frac{n}{2\kappa_1^2}\log\frac{2p}{\epsilon}}\right)\right)$$
$$\le \epsilon + \delta. \qquad \blacksquare$$

## D    Proof for full conformal

**Lemma 7** *Under Assumptions 1 and 2, we have*

$$\mathbb{P}\left(\left\|\mu_{\hat{\beta}_n} - \mu_{\overline{\beta}_n}\right\|_\infty \ge \epsilon\right) \le 2p\exp\left(-\frac{2\kappa_1^2\epsilon^2}{n\kappa_2^2 c_n^2}\right).$$

**Proof:** According to Lemma 5, we have $\|\hat{\beta}_n - \overline{\beta}_n\|_\infty < \epsilon$ with probability at least $1 - 2p\exp\left(-\frac{2\kappa_1^2\epsilon^2}{nc_n^2}\right)$. It follows from Assumption 2 that,

$$\mathbb{P}\left(\left\|\mu_{\hat{\beta}_n} - \mu_{\overline{\beta}_n}\right\|_\infty \ge \epsilon\right) \le \mathbb{P}\left(\kappa_2\left\|\hat{\beta}_n - \overline{\beta}_n\right\|_\infty \ge \epsilon\right) \le 2p\exp\left(-\frac{2\kappa_1^2\epsilon^2}{n\kappa_2^2 c_n^2}\right). \qquad \blacksquare$$

We need the following notation for the proof of Theorem 5:

$$\hat{\beta}_{X_{\text{test}},y} := T(((X_1,Y_1),\ldots,(X_n,Y_n),(X_{\text{test}},y))).$$

### D.1    Proof of Theorem 5

We note,

$$\hat{\mathcal{C}}_\alpha^{\text{full}}(X_{\text{test}}) \supseteq \left\{y \in \mathbb{R}: \frac{1}{n}\sum_{i=1}^n \mathbf{1}\left\{\left|Y_i - \mu_{\hat{\beta}_{X_{\text{test}},y}}(X_i)\right| \ge \left|y - \mu_{\hat{\beta}_{X_{\text{test}},y}}(X_{\text{test}})\right|\right\} > \alpha\right\}$$

$$\supseteq \left\{y \in \mathbb{R}: \frac{1}{n}\sum_{i=1}^n \mathbf{1}\left\{\left|Y_i - \mu_{\hat{\beta}_n}(X_i)\right| - \left|\mu_{\hat{\beta}_n}(X_i) - \mu_{\hat{\beta}_{X_{\text{test}},y}}(X_i)\right| \ge\right.\right.$$

$$\left.\left.\left|y - \mu_{\hat{\beta}_n}(X_{\text{test}})\right| + \left|\mu_{\hat{\beta}_n}(X_{\text{test}}) - \mu_{\hat{\beta}_{X_{\text{test}},y}}(X_{\text{test}})\right|\right\} > \alpha\right\}$$

$$\supseteq \left\{y \in \mathbb{R}: \frac{1}{n}\sum_{i=1}^n \mathbf{1}\left\{\left|Y_i - \mu_{\hat{\beta}_n}(X_i)\right| \ge \left|y - \mu_{\hat{\beta}_n}(X_{\text{test}})\right| + c_{n+1}\right\} > \alpha\right\},$$

where the first and last relations hold according to the definition of $\hat{\mathcal{C}}_\alpha^{\text{full}}(X_{\text{test}})$ and Assumption 1. Assuming $\|\mu_{\hat{\beta}_n} - \mu_{\overline{\beta}_n}\|_\infty < \epsilon$, we obtain

$$\hat{\mathcal{C}}_\alpha^{\text{full}}(X_{\text{test}}) \supseteq \left\{ y \in \mathbb{R} : \frac{1}{n}\sum_{i=1}^n \mathbf{1}\left\{ \left|Y_i - \mu_{\overline{\beta}_n}(X_i)\right| \geq \left|y - \mu_{\overline{\beta}_n}(X_{\text{test}})\right| + c_{n+1} + 2\epsilon \right\} > \alpha \right\}$$

$$\supseteq \left\{ y \in \mathbb{R} : 1 - \hat{F}^{(n)}\left( \left|y - \mu_{\overline{\beta}_n}(X_{\text{test}})\right| + c_{n+1} + 2\epsilon \right) > \alpha \right\}.$$

Assuming $\left\|\hat{F}^{(n)} - F^{(n)}\right\|_\infty < \delta$, we obtain

$$\hat{\mathcal{C}}_\alpha^{\text{full}}(X_{\text{test}}) \supseteq \left\{ y \in \mathbb{R} : 1 - F^{(n)}\left( \left|y - \mu_{\overline{\beta}_n}(X_{\text{test}})\right| + c_{n+1} + 2\epsilon \right) > \alpha + \delta \right\}$$

$$\supseteq \left\{ y \in \mathbb{R} : 1 - F^{(n)}\left( \left|y - \mu_{\overline{\beta}_n}(X_{\text{test}})\right| \right) > \alpha + \delta + (2\epsilon + c_{n+1})L_n \right\}.$$

Therefore,

$$P_e(\mathbf{D}_n) = \mathbb{P}(Y_{\text{test}} \notin \hat{\mathcal{C}}_\alpha^{\text{full}}(X_{\text{test}})|\mathbf{D}_n)$$

$$\leq \mathbb{P}\left( 1 - F^{(n)}\left( \left|Y_{\text{test}} - \mu_{\overline{\beta}_n}(X_{\text{test}})\right| \right) \leq \alpha + \delta + (2\epsilon + c_{n+1})L_n \right)$$

$$= \alpha + \delta + (2\epsilon + c_{n+1})L_n$$

for $\mathbf{D}_n \in \mathcal{A} \cap \mathcal{B}$ where $\mathcal{A} := \left\{ D : \|\mu_{\hat{\beta}_n} - \mu_{\overline{\beta}_n}\|_\infty < \epsilon \right\}$ and $\mathcal{B} := \left\{ D : \left\|\hat{F}^{(n)} - F^{(n)}\right\|_\infty < \delta \right\}$. From Lemma 6, we know $\mathbb{P}(\mathbf{D}_n \notin \mathcal{A}) \leq 2p\exp\left(-\frac{2\kappa_1^2\epsilon^2}{n\kappa_2^2 c_n^2}\right)$. Also, according to Dvoretzky–Kiefer–Wolfowitz inequality, we have $\mathbb{P}(\mathbf{D}_n \notin \mathcal{B}) \leq 2e^{-2n\delta^2}$. Thus,

$$\mathbb{P}(P_e(\mathbf{D}_n) > \alpha + \delta + \epsilon) \leq \mathbb{P}((\mathcal{A} \cap \mathcal{B})^c) \leq 2e^{-2n\delta^2} + 2p\exp\left( -\left( \frac{\kappa_1(\epsilon/L_n - c_{n+1})}{\sqrt{2n}\kappa_2 c_n} \right)^2 \right),$$

or equivalently,

$$\mathbb{P}\left( P_e(\mathbf{D}_n) > \alpha + \sqrt{\frac{\log(2/\delta)}{2n}} + L_n\left( c_{n+1} + \sqrt{2n\log\frac{2p}{\epsilon}}\,\frac{\kappa_2\,c_n}{\kappa_1} \right) \right) \leq \epsilon + \delta. \quad \blacksquare$$

## D.2 Proof of Theorem 6

We note,

$$\hat{\mathcal{C}}_\alpha^{\text{full}}(X_{\text{test}}) = \left\{ y \in \mathbb{R} : \sum_{i=1}^n w_i \mathbf{1}\left\{ \left|Y_i - \mu_{\hat{\beta}_{X_{\text{test}},y}}(X_i)\right| < \left|y - \mu_{\hat{\beta}_{X_{\text{test}},y}}(X_{\text{test}})\right| \right\} < 1 - \alpha \right\}$$

$$\supseteq \left\{ y \in \mathbb{R} : \sum_{i=1}^n \hat{w}_i \mathbf{1}\left\{ \left|Y_i - \mu_{\hat{\beta}_{X_{\text{test}},y}}(X_i)\right| < \left|y - \mu_{\hat{\beta}_{X_{\text{test}},y}}(X_{\text{test}})\right| \right\} < 1 - \alpha \right\}$$

$$= \left\{ y \in \mathbb{R} : \sum_{i=1}^n \hat{w}_i \mathbf{1}\left\{ \left|Y_i - \mu_{\hat{\beta}_{X_{\text{test}},y}}(X_i)\right| \geq \left|y - \mu_{\hat{\beta}_{X_{\text{test}},y}}(X_{\text{test}})\right| \right\} > \alpha \right\}$$

$$\supseteq \left\{ y \in \mathbb{R} : \sum_{i=1}^n \hat{w}_i \mathbf{1}\left\{ \left|Y_i - \mu_{\hat{\beta}_n}(X_i)\right| - \left|\mu_{\hat{\beta}_n}(X_i) - \mu_{\hat{\beta}_{X_{\text{test}},y}}(X_i)\right| \geq \right.\right.$$

$$\left.\left. \left|y - \mu_{\hat{\beta}_n}(X_{\text{test}})\right| + \left|\mu_{\hat{\beta}_n}(X_{\text{test}}) - \mu_{\hat{\beta}_{X_{\text{test}},y}}(X_{\text{test}})\right| \right\} > \alpha \right\}$$

$$\supseteq \left\{ y \in \mathbb{R} : \sum_{i=1}^{n} \hat{w}_i \mathbf{1}\left\{ \left| Y_i - \mu_{\hat{\beta}_n}(X_i) \right| \geq \left| y - \mu_{\hat{\beta}_n}(X_{\text{test}}) \right| + c_{n+1} \right\} > \alpha \right\},$$

where the first and last relations hold according to the definition of $\hat{\mathcal{C}}_\alpha^{\text{full}}(X_{\text{test}})$ (under covariate shift) and Assumption 1. Assuming $\|\mu_{\hat{\beta}_n} - \mu_{\overline{\beta}_n}\|_\infty < \epsilon$, we obtain

$$\hat{\mathcal{C}}_\alpha^{\text{full}}(X_{\text{test}}) \supseteq \left\{ y \in \mathbb{R} : \sum_{i=1}^{n} \hat{w}_i \mathbf{1}\left\{ \left| Y_i - \mu_{\overline{\beta}_n}(X_i) \right| \geq \left| y - \mu_{\overline{\beta}_n}(X_{\text{test}}) \right| + c_{n+1} + 2\epsilon \right\} > \alpha \right\}$$

$$\supseteq \left\{ y \in \mathbb{R} : 1 - \hat{F}_Q^{(n)}\left( \left| y - \mu_{\overline{\beta}_n}(X_{\text{test}}) \right| + c_{n+1} + 2\epsilon \right) > \alpha \right\}.$$

Assuming $\left\|\hat{F}_Q^{(n)} - F_Q^{(n)}\right\|_\infty < \delta$, we obtain

$$\hat{\mathcal{C}}_\alpha^{\text{full}}(X_{\text{test}}) \supseteq \left\{ y \in \mathbb{R} : 1 - F_Q^{(n)}\left( \left| y - \mu_{\overline{\beta}_n}(X_{\text{test}}) \right| + c_{n+1} + 2\epsilon \right) > \alpha + \delta \right\}$$

$$\supseteq \left\{ y \in \mathbb{R} : 1 - F_Q^{(n)}\left( \left| y - \mu_{\overline{\beta}_n}(X_{\text{test}}) \right| \right) > \alpha + \delta + (2\epsilon + c_{n+1})L_{Q,n} \right\}.$$

Therefore,

$$P_e(\mathbf{D}_n) = \mathbb{P}(Y_{\text{test}} \notin \hat{\mathcal{C}}_\alpha^{\text{full}}(X_{\text{test}})|\mathbf{D}_n)$$

$$\leq \mathbb{P}\left( 1 - F_Q^{(n)}\left( \left| Y_{\text{test}} - \mu_{\overline{\beta}_n}(X_{\text{test}}) \right| \right) \leq \alpha + \delta + (2\epsilon + c_{n+1})L_{Q,n} \right)$$

$$= \alpha + \delta + (2\epsilon + c_{n+1})L_{Q,n}$$

for $\mathbf{D}_n \in \mathcal{A} \cap \mathcal{B}(\delta)$ where $\mathcal{A} := \left\{ D : \|\mu_{\hat{\beta}_n} - \mu_{\overline{\beta}_n}\|_\infty < \epsilon \right\}$ and $\mathcal{B}(\delta) := \left\{ D : \left\|\hat{F}_Q^{(n)} - F_Q^{(n)}\right\|_\infty \leq \delta \right\}$. From Lemma 6, we know $\mathbb{P}(\mathbf{D}_n \notin \mathcal{A}) \leq 2p \exp\left( -\frac{2\kappa_1^2 \epsilon^2}{n\kappa_2^2 c_n^2} \right)$. Also, according to weighted DKW inequality from Lemma 1, we have

$$\mathbb{P}\left( \mathbf{D}_n \notin \mathcal{B}\left( \delta + 2C\sqrt{\frac{B}{n}} \right) \right) \leq 4e^{-n\delta^2/(2B^2)}.$$

Thus,

$$\mathbb{P}\left( P_e(\mathbf{D}_n) > \alpha + \delta + 2C\sqrt{\frac{B}{n}} + \epsilon \right) \leq 4e^{-n\delta^2/(2B^2)} + 2p \exp\left( -\left( \frac{\kappa_1(\epsilon/L_{Q,n} - c_{n+1})}{\sqrt{2n}\kappa_2 c_n} \right)^2 \right),$$

or equivalently,

$$\mathbb{P}\left( P_e(\mathbf{D}_n) > \alpha + \left( \sqrt{2B \log 4/\delta} + 2C \right)\sqrt{\frac{B}{n}} + L_{Q,n}\left( c_{n+1} + \sqrt{2n \log \frac{2p}{\epsilon}} \frac{\kappa_2 c_n}{\kappa_1} \right) \right) \leq \epsilon + \delta.$$

∎

# E  Experiment Details for Figure 1

The sizes of the training, calibration, and test datasets are 525, 225, and 247, respectively, for the Wine Quality, Abalone, and Combined Cycle Power Plant datasets. For the Concrete Compressive Strength dataset, the corresponding sizes are 360, 155, and 169. Distribution shift is introduced following the approach of Tibshirani et al. (2019), by resampling data points with probabilities proportional to $\exp(x^\top \beta)$, where

$$\beta_{\text{wine quality}} = [0.5, 1, 0, 0, 0, 0, 0, 0, 0, 0, 0],$$

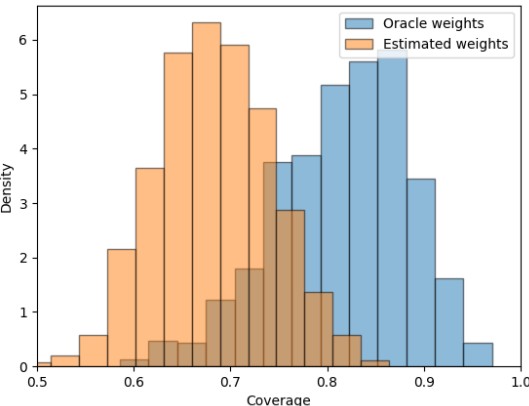

Figure 2: The likelihood ratio is estimated using kernel density estimator with Gaussian kernel.

$$\beta_{\text{abalone}} = [0, 10, 10, 0, 0, 0, 0, 0],$$
$$\beta_{\text{CCS}} = [0.01, 0.01, 0, 0, 0, 0, 0, 0],$$
$$\beta_{\text{CCPP}} = [0.2, 0, 0, 0].$$

As a simple illustration of the role of likelihood ratio estimation, as discussed in Section 4, Figure 2 shows the impact of likelihood ratio estimation on the distribution of the training-conditional error for the Concrete Compressive Strength dataset.

