# OpenReview forum: "Training-Conditional Coverage Bounds under Covariate Shift"
_TMLR — Accepted by TMLR_

### Review · Reviewer_UBba · 2025-08-21

**Summary Of Contributions:**

Conformal prediction is a framework for constructing distribution-free and model-agnostic predictive confidence regions under the exchangeability assumption for the training and test samples; such guarantees are more powerful when they are training-conditional training-conditional guarantees (Bian & Barber, 2023; Liang & Barber, 2023). Another, related line of research (Tibshirani et al., 2019; Prinster et al., 2022) developed "weighted conformal prediction" methods which adapt standard conformal prediction to situations with distribution or covariate shift and guarantee correct marginal coverage. The re-weighting scheme is clever but the fix isn't free. By giving more importance to certain data points in the calibration set, it makes the performance for any specific training set less reliable or "heavier tailed". Instead of just noting that this problem exists, as prior work has done, the authors compute upper bounds on the tails of this training-conditional coverage distribution and use these bounds to quantify the relationship between how bad the tails get and how severe the distribution shift is.

**Audience:**

Yes

**Audience Explanation:**

While there exist foundational papers on training-conditional guarantees and weighted conformal prediction, the specific combination of deriving **training-conditional concentration bounds for weighted conformal methods under algorithmic stability assumptions* is, to the best of my knowledge, unique to this work. It is also significant, not only from a theoretical standpoint (although that would be sufficient) but from a practical standpoint. The bounds presented by the authors in 3.1 are extremely sobering; even with relatively conservative values of B, the bound would quickly become vacuous in practice.

**Claims And Evidence:**

Yes

**Claims Explanation:**

The authors appear to be extremely competent and well-versed in their subject area. Their related works are helpful and thorough, and are cited throughout the paper. The theorems appear to be supported by robust and extensive proofs -- on this point, however, I am prepared to defer to other reviewers who are more active in this subfield than I am, should their opinion differ from my own. In short, I am convinced both that the bounds presented here are not anticipated by prior work, and that they are correct.

**Requested Changes:**

I implore the authors to consider ways to make this work more approachable. The paper lacks a conclusion section, a layman's overview for the discussion section, and any significant experimental results in the main body. While I do not consider any of these failings disqualifying, the absence of these sections impacts the clarity of the evidence presentation, and makes it harder to convince readers who are not already fully immersed in the specifics of this subfield that this result is important.

* Nitpick: they often reduces the concentration -> they often reduce the concentration

---

> ### Author Response · Authors · 2025-10-03
> **Response to Reviewer UBba**
>
> Thank you for supporting our manuscript. We have improved the presentation and address your comments in the revised version.
>
>
> "The paper lacks a conclusion section, a layman's overview for the discussion section, and any significant experimental results in the main body."
>
> Response: Thank you for your suggestions. Our work is mainly theoretical, so our experiments are summarized in Introduction. We have now added a conclusion section that also includes some potential directions in a boarder context.

---

### Review · Reviewer_ZWFG · 2025-08-22

**Summary Of Contributions:**

This paper mainly focuses on conformal prediction. The authors introduces the first training-conditional coverage bounds for conformal prediction under covariate shift, extending beyond marginal coverage guarantees. They derives PAC-style bounds for split and full conformal prediction and jackknife methods. A novel weighted DKW inequalities is proposed to handle non-exchangeable data under covariate shift. They have also conducted experiments to demonstrate the problem of degraded training-conditional coverage under covariate shift using UCI datasets

**Audience:**

Yes

**Audience Explanation:**

This paper tackles ​distribution shift—a fundamental challenge for real-world deployment. It provides theoretical guarantees for uncertainty quantification (conformal prediction), directly supporting ​robustness and safety.

**Claims And Evidence:**

Yes

**Claims Explanation:**

Strengths:
1. This paper is solving practical and essential problems in conformal prediction.
2. The proposed method enjoys better sample dependence than (m,n)-stability baselines.

Weaknesses:
1. The method requires known likelihood ratio dQ/dP. Uniform stability only holds for simple models.
2. The given bounds depend on unknown constants (κ1, κ2, Ln) and stability parameters (cn). The presentation can be improved.
3. Impact of likelihood ratio estimation (Sec. 4) or stability parameters not tested.
4. It would be better to conduct experiments on some real-work data benchmarks.

**Requested Changes:**

Please refer to the weaknesses.

---

> ### Author Response · Authors · 2025-10-03
> **Response to Reviewer ZWFG**
>
> Thank you for your insightful comments.
>
> 1. Thank you for pointing this out. Please note that we have included the impact of the unknown likelihood ratio in the discussion section. The result for split conformal does not require uniform stability.
>
> 2. The current result for jackknife+ and full-conformal prediction is more of theoretical value. We have improved the presentation in the revised version.
>
> 3. We have added Figure 2 to show the impact of the likelihood ratio estimation to the experiments in the revised version.
>
> 4. The experiments in the introduction section include four popular real-world datasets.

---

### Review · Reviewer_n8tU · 2025-09-19

**Summary Of Contributions:**

The paper establishes data set conditional guarantees on the confidence intervals obtained by several techniques, including jackknife/cross-validation.

**Audience:**

Yes

**Audience Explanation:**

The paper considers some well-known methods for obtaining estimates of parameters and derives some new confidence bounds for these. I believe this could be of reasonably broad interest.

**Broader Impact Concerns:**

No concerns. This is a theoretical work.

**Claims And Evidence:**

No

**Claims Explanation:**

The paper describes proofs of the claims, but there are a few points where I could not verify the proofs. The most significant is in the proof of Theorem 1, at the top of p.18. There it is claimed that conditioned on the dataset, the probability of a value falling outside a quantile of the true distribution function is at most the probability that the value falls outside a quantile of the empirical distribution function. I don't see why this should be true. Later theorems include an analysis of the difference between the empirical and true distribution functions, so I imagine that this can be addressed, but it might involve some additional loss in the bounds.

In the other places, the presentation is simply incomplete. The most significant example of this is in the proof of Lemma 4 (p.17) that references the proof of a theorem in another work. Since this is the appendix of a journal submission, length is not at issue, and I would have expected to see the relevant portion of the argument here. (Naturally, the authors should still be acknowledged.) Without the full argument, I can't verify it. In other cases, the proof relies on a theorem proved elsewhere and for the sake of being self-contained, I would have expected to at least see the statement of the claim (not the proof in this case) from those works reproduced here.

**Requested Changes:**

It is critical that the gap in Theorem 1 is addressed, and that the relevant argument in Lemma 4 is included. The statements of the several other theorems that are used, in Lemma 1, Lemma 3, and Theorem 3.

There are also several points in the main paper where the notation is not clear. Section 4 discusses "(m,n)-stability" without explaining what it is: I don't know what the psi functions in equations 9 and 10 mean. In assumption 3, I don't entirely understand the notation -- why is a prime attached to F (and not to F^{(n)}, which is what I would expect if it was a derivative bound). If this is simply a matter of ugly notation, the assumption seems to come down to the function being Lipschitz, and perhaps it should just be explained in those terms. It was not clear if the bounds here are simply constants. In Theorem 2, there is an expression (dQ/dP)^2(Z_test), and it is not clear if this means that the output is squared or if the function is iterated.

The rest are less critical, but still important:

The error function for Jackknife+ is used in Theorems 3 & 4 without being defined (and similarly for CV+ in Corollary 2, etc.). The definition should be included. Relatedly,  I assume that data-conditional probability of error in these functions is w.r.t. Q, but please clarify this.

In remark 1, it is stated that the function should be continuously differentiable for some open set that contains the entire parameter space. Is this what was intended? (If so, can you include an example?)

I can guess what you mean by ||.||_{P,2} in corollary 1, but I would not advise using this notation. At minimum, it should be recalled. Incidentally, a significant portion of the audience won't be familiar with absolute continuity and won't be able to make sense of the notation "<<" in Theorem 1. (I don't think you could even google this.) It isn't clear that you even need to state it since you're already assuming the derivative exists.

It would also be helpful to recall the definition of the bracketing number in the appendix before you start reasoning about it, at minimum for the sake of fixing the meaning of the notation.

---

> ### Author Response · Authors · 2025-10-03
> **Response to Reviewer n8tU**
>
> Thank you for your detailed comments and suggestions.
>
>
> "It is critical that the gap in Theorem 1 is addressed, and that the relevant argument in Lemma 4 is included. The statements of the several other theorems that are used, in Lemma 1, Lemma 3, and Theorem 3."
>
> Response: In the proof of Theorem 1, at the top of p.18 (now 19), please note that $F_{Q,m}$ defined in (25) is not the true distribution function. In particular, $F_{Q,m}$ and $\hat {F}_{Q,m}$ denote the empirical weighted distribution function and its modified version, respectively. The statement of the used theorems are now added to the manuscript.
>
>
>
> Notation issues: "psi functions in equations 9 and 10", "prime attached to F", "expression $(dQ/dP)^2(Z_{test})$", " $||.||_{P,2}$ in corollary 1", "explain $\ll$".
>
> Response: Thank you for pointing out the presentation and notation issues. We have improved the presentation and addressed your comments in the revised version.
>
>
>
> "The error function for Jackknife+ is used in Theorems 3 and 4 without being defined (and similarly for CV+ in Corollary 2, etc.)"
>
> Response: The error functions are now defined explicitly.
>
>
> Clarification: "I assume that data-conditional probability of error in these functions is w.r.t. Q"
>
> Response: It is explained in the statement of Theorem 1.
>
>
>
> Clarification: "In remark 1, it is stated that the function should be continuously differentiable for some open set that contains the entire parameter space. Is this what was intended?"
>
> Response:
> Yes, this is correct. The linear model is an example.
>
>
> Clarification: "recall the definition of the bracketing number in the appendix before you start reasoning about it"
>
> Response: It is now defined in Appendix A.

---

### Comment · Editors_In_Chief · 2026-01-30

On 1/29/2026, by the request of the authors, the EiCs replaced the camera ready revision with a version that correctly indicates the publication date.

---

### Decision · Action_Editor_gMW3 · 2025-12-04

**Recommendation:** Accept with minor revision

**Additional Comments:**

Reviewer n8tU recommendation regarding theorems 9 and 10 should be addressed.

**Audience:**

Yes

**Audience Explanation:**

Conformal prediction under covariate shift is a standard problem that comes in many settings.

**Claims And Evidence:**

Yes

**Claims Explanation:**

This paper solves practical and essential problems in conformal prediction. It introduces the first training-conditional coverage bounds for conformal prediction under covariate shift, which extend beyond marginal coverage guarantees. The paper derives PAC-style bounds for split and full conformal prediction, as well as jackknife methods. A novel weighted DKW inequality is proposed to handle non-exchangeable data under covariate shift.   The proposed method has better sample dependence than (m, n)-stability baselines.
Theoretical results are backed up by practical experiments.